



# Development of the global atmospheric general circulation-chemistry model BCC-GEOS-Chem v1.0: model description and evaluation

Xiao Lu[1,2], Lin Zhang[1,*], Tongwen Wu[3,*], Michael S. Long[2], Jun Wang[4], Daniel J. Jacob[2], Fang Zhang[3],

Jie Zhang[3], Sebastian D. Eastham[5], Lu Hu[6], Lei Zhu[2,7,8], Xiong Liu[7], and Min Wei[9]

(1) Laboratory for Climate and Ocean-Atmosphere Sciences, Department of Atmospheric and Oceanic Sciences, School of Physics, Peking University, Beijing 100871, China

(2) School of Engineering and Applied Sciences, Harvard University, Cambridge, MA 02138, USA

(3) Beijing Climate Center, China Meteorological Administration, Beijing 100081, China

(4) University of Iowa, Iowa City, IA 52242, USA

(5) Laboratory for Aviation and the Environment, Massachusetts Institute of Technology, Cambridge, MA 02139, USA

(6) Department of Chemistry and Biochemistry, University of Montana, Missoula, MT 59812, USA

(7) Harvard-Smithsonian Center for Astrophysics, Cambridge, MA 02138, USA

(8) School of Environmental Science and Engineering, Southern University of Science and Technology, Shenzhen 518055, Guangdong, China

(9) National Meteorological Information Center, China Meteorological Administration, Beijing 100871, China

*Correspondence to* Lin Zhang (zhanglg@pku.edu.cn) and Tongwen Wu (twwu@cma.gov.cn)

**Abstract**

Chemistry plays an indispensable role in investigations of the atmosphere, however, many climate models either ignore or greatly simplify atmospheric chemistry, limiting both their accuracy and their scope. We present the development and evaluation of the online global atmospheric chemical model BCC-GEOS-Chem v1.0, coupling the GEOS-Chem chemical transport model (CTM) as an atmospheric chemistry component in the Beijing Climate Center atmospheric

general circulation model (BCC-AGCM). The GEOS-Chem atmospheric chemistry component includes detailed tropospheric $HO_x$-$NO_x$-VOC-ozone-bromine-aerosol chemistry and online dry and wet deposition schemes. We then demonstrate the new capabilities of BCC-GEOS-Chem v1.0 relative to the base BCC-AGCM model through a three-



year (2012-2014) simulation with anthropogenic emissions from the Community Emissions Data System (CEDS) used
in the Coupled Model Intercomparison Project Phase 6 (CMIP6). The model well captures the spatial distributions and
seasonal variations in tropospheric ozone, with seasonal mean biases of 0.4-2.2 ppbv at 700-400 hPa compared to
satellite observations and within 10 ppbv at the surface-500 hPa compared to global ozonesonde observations. The
model has larger high ozone biases over the tropics which we attribute to an overestimate of ozone chemical production.
It underestimates ozone in the upper troposphere which likely due to either the use of a simplified stratospheric ozone
scheme and/or to biases in estimated stratosphere-troposphere exchange dynamics. The model diagnoses the global
tropospheric ozone burden, OH concentration, and methane chemical lifetime to be 336 Tg, $1.16\times10^6$ molecule $cm^{-3}$,
and 8.3 years, respectively, consistent with recent multi-model assessments. The spatiotemporal distributions of $NO_2$,
CO, $SO_2$, $CH_2O$, and aerosols optical depth are generally in agreement with satellite observations. The development of
BCC-GEOS-Chem v1.0 represents an important step for the development of fully coupled earth system models (ESMs)
in China.




## 1. Introduction

Atmospheric chemistry plays an indispensable role in the evolution of atmospheric gases and aerosols, and is also an essential component of the climate system due to its active interactions with atmospheric physics and biogeochemistry on various spatiotemporal scales. Climate modulates the natural emissions, chemical kinetics, and transport of atmospheric gases and aerosols, while changes in many of these constituents alter the radiative budgets of the climate system and also influence the biosphere (Jacob and Winner, 2009; Fiore et al., 2012; Lu et al., 2019). Climate-chemistry coupled models are indispensable tools to quantify climate-chemistry interactions and to predict future air quality. However, coupling transport and chemistry of hundreds of chemical species on all spatiotemporal scales in climate system models (CSMs) posts a considerable challenge for model complexity and computational resources. Only ten of the thirty-nine CSMs in the phase five of the Coupled Model Intercomparison Project (CMIP5) simulated atmospheric chemistry interactively (IPCC AR5, 2013). In the other models, including all the five Chinese CSMs, chemically active species were prescribed. Development of climate-chemistry coupled model has been identified as a research frontier for atmospheric chemistry (National Research Council, 2012), and also a priority for CSM development particularly in China. An initiative to include online simulation of atmospheric chemistry in CSM, as an essential step toward building a climate-chemistry coupled model, was launched in the Beijing Climate Center (BCC) at the China Meteorological Administration (CMA) after CMIP5.

Here we present the development of the global atmospheric chemistry-general circulation model BCC-GEOS-Chem v1.0, which enables online simulation of atmospheric chemistry in the BCC-CSM version 2 (BCC-CSM2). BCC-GEOS-Chem is built on the coupling of the GEOS-Chem chemical module to the BCC Atmospheric General Circulation Model (BCC-AGCM), the atmospheric component of the BCC-CSM2. BCC-CSM2 is a fully coupled global CSM in which the atmosphere, land, ocean, and sea-ice components interact with each other through the exchange of momentum, energy, water, and carbon (Wu et al., 2008, 2013, 2019). The earlier version of BCC-CSM (v1.1 and v1.1m) was enrolled in CMIP5 and has been widely applied on weather and climate research (e.g., Wu et al., 2013, 2014; Xin et al., 2013; Zhao and He, 2015). It has been recently updated to BCC-CSM2 and is being used in that configuration for the Coupled Model Intercomparison Project Phase 6 (CMIP6) (Wu et al., 2019). GEOS-Chem (http://geos-chem.org), originally described by Bey et al. (2001) is a global three-dimensional chemical transport model (CTM) which includes detailed state-of-science gas-aerosol chemistry, is used by a large international community for a broad range of research on atmospheric chemistry, is continually updated with scientific innovations from users, is rigorously benchmarked, and is openly accessible on the cloud (Zhuang et al., 2019). The model is continually evaluated with atmospheric observations by its user community (e.g., Hu et al., 2017). The integration of BCC-AGCM and GEOS-Chem for online simulation of global atmospheric chemistry will have attractive scientific and operational applications (e.g., sub-seasonal air quality



prediction), and also represents an important step for the development of fully coupled earth system models (ESMs) in
China.

Until recently, the offline GEOS-Chem CTM relied exclusively on fixed longitude-latitude grids and was designed for
shared-memory (OpenMP) parallelization. With such features the GEOS-Chem CTM was not flexible to be coupled
with BCC-CSM, which typically runs on spectral space (with adjustable options for the grid type and resolution
dependent on the wave truncation number) and requires vast computational resources. Integration of GEOS-Chem
chemical module into CSMs has been enabled by separating the module (which simulates all local processes including
chemistry, deposition, and emission) from the simulation of transport, and making it operate on 1-D (vertical) columns
in a grid-independent manner (Long et al., 2015; Eastham et al., 2018). The GEOS-Chem chemical module can thus be
coupled with a CSM on any grid, and the CSM simulation of dynamics then handles chemical transport. GEOS-Chem
used as an online chemical module in CSMs shares the exact same code as the classic offline GEOS-Chem for local
processes (chemistry, deposition, and emission) (Long et al., 2015). This capability ensures that the scientific
improvements of GEOS-Chem contributed from worldwide research community can be conveniently incorporated into
CSMs, allowing the chemistry of BCC-GEOS-Chem to be trackable to the latest GEOS-Chem version. Previous studies
have demonstrated the success of coupling GEOS-Chem into the NASA GEOS-5 Earth system model as an online
atmospheric chemistry module (Long et al., 2015; Hu et al., 2018).

This paper presents the overview of the BCC-GEOS-Chem v1.0 model, and evaluates the model simulation of present-
day atmospheric chemistry. The model framework and its components are described in Section 2. We conducted a three-
year (2012-2014) model simulation to demonstrate the model capability and for model evaluation. In section 3, we
compare simulated gases and aerosols with satellite and *in-situ* observations, and also diagnose the global tropospheric
ozone burden and budget. Future plans for model development and summary are presented in Section 4.

## 2. Development and description of the BCC-GEOS-Chem v1.0

Figure 1 presents the framework of the BCC-GEOS-Chem v1.0. BCC-GEOS-Chem v1.0 includes interactive
atmosphere (including dynamics, physics, and chemistry) and land modules, and other components such as ocean and
sea ice are configured as boundary conditions for this version. Atmospheric dynamics and physics module *(Section 2.1)*
and the land module *(Section 2.2)* come from the BCC-AGCM version 3 and the BCC Atmosphere and Vegetation
Interaction Model version 2 (BCC-AVIM2), respectively. Atmosphere and land modules exchange the fluxes of
momentum, energy, water, and carbon through the National Center for Atmospheric Research (NCAR) flux Coupler
version 5. Dynamic and physical parameters from both the atmosphere (e.g., radiation, temperature, and wind) and the



land modules (e.g., surface stress and leaf area index) are then used to drive the GEOS-Chem chemistry *(Section 2.3)* and deposition *(Section 2.4)* of atmospheric gases and aerosols. Anthropogenic and biomass burning emissions are from the inventories used for the CMIP6 *(Section 2.5.1)*. A number of climate-sensitive natural emissions such as biogenic and lightning emissions are calculated online in the model (*Section 2.5.2*). Boundary conditions, external forcing, and experiment design are described in *Section 2.6*.

## 110  2.1 The atmospheric model BCC-AGCM3

BCC-AGCM3 is a global atmospheric spectral model. It has adjustable horizontal resolution and 26 vertical hybrid layers extending from the surface to 2.914 hPa. In this study we use the default horizontal spectral resolution of T42 (approximately 2.8° latitude × 2.8° longitude). The dynamical core and physical processes of the BCC-AGCM3 have been described comprehensively in Wu et al. (2008, 2010) with recent updates documented in Wu et al. (2012, 2019).

Wu et al. (2019) showed that the BCC-CSM2 (BCC-AGCM3 as the atmospheric model) well captured the global patterns of temperature, precipitation, and atmospheric energy budget. BCC-CSM2 also showed significant improvements in reproducing the historical changes of global mean surface temperature from 1850s and climate variabilities such the quasi-biennial oscillation (QBO) and the El Niño–Southern Oscillation (ENSO) compared with its previous version BCC-CSM1.1m (Wu et al., 2019). Here we present a brief summary of the main features in BCC-

AGCM3.

The governing equations and physical processes (e.g., clouds, precipitation, radiative transfer, and turbulent mixing) of BCC-AGCM3 are originated from the Eulerian dynamic framework of the Community Atmosphere Model (CAM3) (Collins et al., 2006), but substantial modifications have been incorporated. Wu et al. (2008) introduced a stratified

reference of atmospheric temperature and surface pressure to the governing equations. In this way, prognostic temperature and surface pressure in the original governing equation can be derived from their prescribed reference plus the prognostic perturbations relative to the reference. Resolving algorithms (e.g., explicit and semi-implicit time difference scheme) were adapted accordingly. The modified dynamic framework reduced the truncation errors in the model as well as the bias due to inhomogeneous vertical stratification, and therefore improved the descriptions of the

pressure gradient force and the vertical temperature structure (Wu et al., 2008). BCC-AGCM3 also implements a new mass-flux cumulus scheme to parameterize deep convection (Wu, 2012). The revised deep convection parameterization by including the entrainment of environment air into the uplifting parcel better captured the realistic timing of intense precipitation (Wu, 2012) and the Madden-Julian Oscillation (MJO) (Wu et al., 2019). Other important updates of atmospheric physical processes in BCC-AGCM3 relative to CAM3 include a new dry adiabatic adjustment to conserve

the potential temperature, a modified turbulent flux parameterization to involve the effect from waves and sea spray on





ocean surface latent and sensible heat, a new scheme to diagnose cloud fraction, a revised cloud microphysics scheme to include the aerosol indirect effects based on bulk aerosol mass, and modifications for radiative transfer and boundary layer parameterizations (Wu et al., 2010; 2014; 2019).

## 2.2 The land model BCC-AVIM2

BCC-AVIM2 is a comprehensive land surface model originated from the Atmospheric and Vegetation Interaction Model (AVIM) (Ji, 1995; Ji et al., 2008), and serves as the land component in BCC-CSM2. It includes three submodules: the biogeophysical module, plant ecophysiological module, and soil carbon-nitrogen dynamic module. The biogeophysical module simulates the transfer of energy, water, and carbon between the atmosphere, plant canopy, and soil. It has 10 soil layers and up to 5 snow layers. The ecophysiological module describes the ecophysiological activities such as

photosynthesis, respiration, turnover, and mortality of vegetation, and diagnoses the induced changes of biomass. The soil carbon-nitrogen dynamic module describes the biogeochemical process such as the conversion and decomposition of soil organic carbon. The vegetation surface in BCC-AVIM2 is divided into 15 plant functional types (PFTs) as shown in Table 1, and each grid cell contains up to 4 PFTs types. Wu et al. (2013) showed that the model well captured the spatial distributions, long-term trends, and interannual variability of global carbon sources and sinks compared to

observations and other models. Recent improvements in BCC-AVIM2, such as the introduction of a variable temperature threshold for the thawing/freezing of soil water, and improved presentations of snow surface albedo and snow cover fraction, are described in Li et al. (2019). Biogenic emissions and dust mobilizations are also implemented in BCC-AVIM2 interactively with the atmosphere, as will be described later in Section 2.5.

## 2.3. Atmospheric chemistry

We implement in this study the GEOS-Chem v11-02b "Tropchem" mechanism as the atmospheric chemistry module of BCC-GEOS-Chem v1.0. As described in the introduction, GEOS-Chem used as an online chemical module in ESMs shares the exact same codes for local terms (chemistry, deposition, and emission) as the classic offline GEOS-Chem. Here, we use the GEOS-Chem chemical module to process chemistry and deposition in BCC-GEOS-Chem v1.0, and operate emission separately in the model as will be described in Section 2.5.


GEOS-Chem v11-02b "Tropchem" mechanism describes advanced and detailed $HO_x$-$NO_x$-VOC-ozone-bromine-aerosol chemistry relevant to the troposphere (Mao et al., 2010, 2013; Parrella et al., 2012; Fischer et al., 2014; Marais et al., 2016). It includes 74 advected species (tracers) and 91 non-advected species (http://wiki.seas.harvard.edu/geos-chem/index.php/Species_in_GEOS-Chem). Tracer advection in BCC-GEOS-Chem v1.0 is performed using a semi-



Lagrangian scheme (Williamson and Rasch, 1989). Photolysis rates are calculated by the Fast-JX scheme (Bian and
Prather, 2002). The simulation of sulfate-nitrate-ammonia (SNA) aerosol chemistry, four-size bins of mineral dust (radii
of 0.1-1.0, 1.0-1.8, 1.8-3.0, and 3.0-6.0 μm), and two types of sea salt aerosols (accumulating mode: 0.01-0.5; coarse
mode: 0.5-8.0 μm) follows Park et al. (2004), Fairlie et al. (2007), and Jaegle et al. (2011). Aerosol and gas-phase
chemistry interacts through heterogeneous chemistry on aerosol surface (Jacob, 2000; Evans and Jacob, 2005; Mao et

al., 2013), aerosol effects on photolysis (Martin et al., 2003), and gas-aerosol partitioning of $NH_3$ and $HNO_3$ calculated
by the ISORROPIA II thermodynamic module (Fontoukis and Nenes, 2007; Pye et al., 2009). Methane concentrations
in the chemistry module are prescribed as uniform mixing ratios over four latitudinal bands (90°–30°S, 30°S–0°, 0°–
30°N, and 30°–90°N), with the year-specific annual mean concentrations given by surface measurements from the
NOAA Global Monitoring Division. Stratospheric ozone is calculated by the linearized ozone parameterization (LINOZ)

(McLinden et al., 2000) and is transported to the troposphere driven by the model wind fields.

### 2.4. Dry and wet deposition

Dry and wet deposition for both gas and aerosols are parameterized following GEOS-Chem algorithms. Dry deposition
is calculated online based on the resistance-in-series scheme (Wesely, 1989). The scheme describes gaseous dry
deposition by three separate processes, *i.e.*, the turbulent transport in aerodynamic layer, molecular diffusion through

the quasi-laminar boundary layer, and uptake at the surface. Aerosol dry deposition further considers the gravitational
settling of particles as described in Zhang et al. (2001). Variables needed for the dry deposition calculation such as the
friction velocity, Monin-Obukhov length, and leaf area index (LAI) are obtained from the atmospheric dynamics/physics
modules or the land module BCC-AVIM. We have also reconciled the land use types (LUT) used in dry deposition with
those used in BCC-AVIM, following Geddes et al. (2016) and Zhao et al. (2017). The LUTs from BCC-AVIM are

mapped directly to the 11 deposition surface types used that in GEOS-Chem as shown in Table 1. Dry deposition velocity
is calculated as the weighted average over all LUTs in each grid box.

Wet deposition of aerosols and soluble gases by precipitation in BCC-GEOS-Chem v1.0 includes the scavenging in
convective updrafts, in-cloud rainout, and below-cloud washout (Liu et al., 2001). Following the implementation of

GEOS-Chem chemical module to GEOS-5 ESM (Hu et al., 2018), convective transport of chemical tracers and
scavenging in the updrafts in BCC-GEOS-Chem v1.0 is performed using the GEOS-Chem convection scheme but with
convection variables diagnosed from BCC-AGCM. This takes advantages of the existing capability of the GEOS-Chem
scheme to describe gas and aerosol scavenging (Liu et al., 2001; Amos et al., 2012).



## 2.5 Emissions

### 2.5.1 Offline emissions

Historical anthropogenic emissions used in this study are obtained from the Community Emissions Data System (CEDS) emission inventory (Hosely et al., 2018). CEDS is an updated global emission inventory which provides sectoral, gridded, and monthly emissions of reactive gases and aerosols from 1750-2014 for use in the CMIP6 experiment (Eyring et al. 2016; Hosely et al., 2018). Here we use the CEDS anthropogenic emissions of $NO_x$, CO, $SO_2$, $NH_3$, non-methane volatile organic compounds (NMVOCs), and carbonaceous aerosols (black carbon (BC) and organic carbon (OC)) (Table 2). We also include three-dimensional aircraft emissions of several gases and aerosols in the model. The historical global biomass burning emission inventory is obtained from van Marle et al. (2017) which is also used for the CMIP6 experiment. Prescribed soil $NO_x$, volcano $SO_2$, and ocean dimethyl sulfide (DMS) emissions from the CMIP5 dataset are also included.

Table 2 lists the amount of annual total emissions of chemicals used in this study separated by emission sectors averaged over 2012-2014. Figure 2a and 2b shows the spatial distributions of annual NO (not including lightning emissions which will be discussed below separately) and CO emissions. Global annual total emissions of NO (not including lightning emissions) and CO are 111.1 and 925.5 Tg year$^{-1}$, respectively. The global anthropogenic emissions are relatively flat in 2012-2014 (e.g. from 614.7 to 619.6 Tg year$^{-1}$ for CO), while the biomass burning emissions have much stronger interannual variability (e.g., varying from 209 to 256 Tg year$^{-1}$ for CO). As pointed out by Hosely et al. (2018), CEDS anthropogenic emissions are generally higher than previous inventories. For instance, the anthropogenic $NO_x$ and CO emissions are, respectively, about 10% and 8% higher compared to CMIP5 emissions in 1980-2000 periods. This is likely due to the updates of emission factors and inclusions of new emission sectors (Hosely et al., 2018). Compared to CMIP5, biomass burning emissions used in CMIP6 for the 2000 condition is about 20% and 30% lower for CO and OC, respectively, but is about 17% higher for $NO_x$ (Fig. 13 in van Marle et al., 2017).

### 2.5.2 Online emissions

BCC-GEOS-Chem v1.0 includes a number of climate-sensitive natural sources. Biogenic emissions of NMVOCs are calculated online using the Model of Gases and Aerosols (MEGAN) algorithm (Guenther et al., 2012) in the land module. MEGAN estimates biogenic emissions as a function of an emission factor at standard condition, a normalized emission activity factor relative to the standard condition, and a scaling ratio which accounts for canopy production and loss. The emission activity factor is further determined by surface or plant parameters such as leaf age and LAI diagnosed in





BCC-AVIM, as well as meteorological variables such as radiation and temperature. The annual biogenic isoprene
225 emissions calculated in BCC-GEOS-Chem v1.0 are 410.0 Tg year$^{-1}$ averaged for 2012-2014 period with a relatively
small interannual variability (404.6 to 415.2 Tg year$^{-1}$). This is close to but lower than estimates from the literature (500-
750 Tg, Guenther et al., 2012). The model captures the hot spots of biogenic isoprene emissions in the tropical continents
and the southeastern US (Fig. 2c).

Parameterization of lightning NO emissions follows Price and Rind (1992). The model diagnoses the lightning flash
frequency in deep convection as a function of the maximum cloud-top-height (CTH). Lightning NO production is then
calculated as a function of lightning flash frequency, fraction of intracloud (IC) and cloud-to-ground (CG) lightning
based on the cloud thickness, and the energy per flash (Price et al., 1997). Vertical distributions of lightning NO
emissions in the column follow Ott et al. (2010). The model estimates t global annual total lighting NO emissions of
10.9 to 12.2 Tg NO year$^{-1}$ for 2012-2014, in agreement with the best estimate of present-day emissions (10.7±6.4 Tg
NO year$^{-1}$ as summarized in Schumann and Huntrieser, 2007). The emissions are centered near the tropics due to strong
convection as shown in Figure 2d.

The model also includes wind-driven sea salt and mineral dust emissions. Emission fluxes of sea salt aerosols are
dependent on the sea salt particle radius and proportional to the 10-meter wind speed with a power of 3.41 following
the empirical parameterization from Monahan et al. (1986) and Gong et al. (1997). Mineral dust emissions are
determined by wind friction speed, soil moisture, and vegetation type following the Dust Entrainment and Deposition
(DEAD) scheme as described by Zender et al. (2003). Figure 2e and 2f show the spatial distributions of sea salt and
mineral dust emissions, with their annual total emissions of 3963 Tg and 1347 Tg, respectively, consistent with previous
estimates from Jaeglé et al. (2011) and Fairlie et al. (2007).

### 2.6 Boundary conditions, external forcing, and experiment design

BCC-GEOS-Chem v1.0 is configured using prescribed ocean and sea ice as boundary conditions. Historical sea surface
temperature and sea-ice extents are obtained from (https://esgf-node.llnl.gov/search/input4mips/, last access: 2 June
2019). These prescribed datasets are also used in CMIP6 atmosphere-only simulations. External forcing data, including
historical greenhouse gas concentrations ($CO_2$, $CH_4$, $N_2O$, CFCs) (Meinshausen et al., 2017), land use forcing, and solar
forcing, are also accessed from (https://esgf-node.llnl.gov/search/input4mips/). BCC-CSM2 has implemented the
radiative transfer effects of greenhouse gases and aerosols as well as the aerosol-cloud interactions based on bulk aerosol
mass concentrations (Wu et al., 2019). Since BCC-CSM2 does not include interactive atmospheric chemistry, the
calculation of radiative transfer and aerosol-cloud interactions are based on historical gridded ozone concentrations from



CMIP5 and CMIP6-recommended anthropogenic aerosol optical properties (Stevens et al., 2017). Here for BCC-GEOS-Chem v1.0, we follow BCC-CSM2 and use these prescribed ozone and aerosols rather than model online calculated values for feedback calculation. This is meant to focus on modeling and evaluation of atmospheric chemistry in this work as the first step of the coupling. Interactive coupling of chemistry and climate through radiation and aerosol-cloud interactions will be considered in the next version of BCC-GEOS-Chem.


We conduct BCC-GEOS-Chem v1.0 simulations from 2011 to 2014. The initial conditions for atmospheric dynamics and physics at 2011 are obtained from the historical simulations (1850-2014) of BCC-CSM2 (Wu et al., 2019), and initial states of chemical tracers are obtained from the GEOS-Chem Unit Tester (http://wiki.seas.harvard.edu/geos-chem/index.php/Unit_Tester_for_GEOS-Chem_12, last access: 2 June 2019). Model results for 2012-2014 are

evaluated.

## 3. Model evaluation

### 3.1 Observations used for model evaluation

We use an ensemble of surface, ozonesonde, and satellite observations to evaluate the BCC-GEOS-Chem v1.0 simulation of present-day atmospheric chemistry (Table 3). Ozonesonde measurements are obtained from the World

Ozone and Ultraviolet Radiation Data Centre (WOUDC; http://woudc.org/data.php, last access: 2 June 2019) operated by the Meteorological Service of Canada. The network also includes sites from the Southern Hemisphere Additional Ozonesondes (SHADOZ, Thompson et al., 2003). To derive the monthly mean ozone profiles, only sites and months with more than three observations in the month are considered. We further categorize the WOUDC observations into nine regions following Tilmes et al. (2012) and Hu et al. (2017) for model evaluation as shown in Figure 3. We also use

the TOAR surface ozone database (Schultz et al., 2017a) that provides ozone metrics (e.g. monthly mean) for more than 9000 monitoring sites around the world from the 1970s to 2014 (Schultz et al., 2017b). Surface aerosol measurements (sulfate, nitrate, OC, BC) over the US are obtained from the Interagency Monitoring of Protected Visual Environments (IMPROVE) network. These aerosol measurements are 24-hour averages every 3 days.

Satellite products from the NASA Earth Observing System (EOS) Aura satellite's Ozone Monitoring Instrument (OMI) are also used. We use the OMI PROFOZ ozone profiles with 24 layers extending from the surface to 60 km retrieved by Liu et al. (2005; 2010) based on the optimal estimation technique (Rodgers, 2000). The OMI PROFOZ dataset has been comprehensively validated by comparisons with ozonesondes (Zhang et al., 2010; Hu et al., 2017; Huang et al., 2017) and satellite products (Huang et al., 2018). We also use the OMI gridded monthly mean tropospheric column of



nitrogen dioxide (NO$_2$) (Krotkov et al., 2013), formaldehyde (CH$_2$O) (De Smedt et al., 2015), and planetary boundary

layer (PBL) sulfur dioxide (SO$_2$) column (Krotkov et al., 2015). Other satellite observations include total column carbon

monoxide (CO) observations from Measurements of Pollution in the Troposphere (MOPITT) (Deeter et al., 2017), and

aerosol optical depth (AOD) at 550 nm from the Moderate Resolution Imaging Spectroradiometer (MODIS) (available

at https://neo.sci.gsfc.nasa.gov/view.php?datasetId=MODAL2_M_AER_OD, last access: 2 June 2019). Satellite

observations are further re-gridded to the model resolution for model evaluation, except for the MODIS AOD dataset

due to a large number of invalid measurements.

**3.2 Evaluation of tropospheric ozone with observations**

Figure 4 shows the spatial and seasonal distributions of mid-tropospheric ozone (700-400 hPa) from OMI satellite

observations and BCC-GEOS-Chem v1.0 simulation averaged over 2012-2014, as well as their differences. We analyze

ozone at 700-400hPa where OMI satellite has the peak sensitivity (Zhang et al., 2010). Model outputs are sampled along

the OMI tracks and smoothed with OMI averaging kernels for proper comparison to the observations (Zhang et al., 2010;

Hu et al., 2017, 2018; Lu et al., 2018).

The model well captures the main features of tropospheric ozone distribution and seasonal variation. Both satellite

observations and BCC-GEOS-Chem v1.0 model results show high mid-tropospheric ozone levels over the northern

mid-latitudes in boreal spring due to stronger stratospheric influences and in summer due to higher photochemical

production, and over the Atlantic and southern Africa during boreal autumn driven by strong biomass burning emissions

(Fig. 2), lightning NOx and dynamical processes (e.g., Sauvage et al., 2007). The spatial patterns of observed and

simulated tropospheric ozone values are highly correlated, with correlation coefficients (*r*) of 0.79-0.93. BCC-GEOS-

Chem v1.0 shows small global seasonal mean biases of 0.4~2.2 ppbv relative to OMI observations, comparable to the

biases of 0.1~2.7 ppbv for G5NR-Chem (NASA GEOS-ESM with GEOS-Chem v10-01 as an online chemical module)

in a similar period (Hu et al., 2018). We find that BCC-GEOS-Chem v1.0 tends to overestimate tropospheric ozone

levels over tropical oceans by 3-12 ppbv and underestimate ozone over the northern mid-latitudes by 3-9 ppbv, similar

to the patterns simulated by the classic GEOS-Chem and G5NR-Chem models (Hu et al., 2017, 2018).


Comparisons with global ozonesonde observations further demonstrate that BCC-GEOS-Chem v1.0 has no significant

biases in the tropospheric ozone simulation. As shown in Figure 5, the model well reproduces the observed ozone

vertical structures, e.g., the slow increase of ozone with increasing altitude in the troposphere, and the sharp ozone

gradient near and above the tropopause. Figure 6 compares seasonal variations of ozone concentrations in different

regions at three tropospheric levels (800 hPa, 500 hPa, and 300 hPa). Overall, the model reproduces the ozone annual



cycles driven by different chemical and dynamical processes. The model captures the springtime and summertime ozone peaks at the northern mid-latitudes (Japan, US, Europe, Canada) ($r$=0.53~0.94 for different layers), but only fairly reproduces the annual ozone cycle in the Southern Hemisphere (SH) and the tropics. Mean model biases at the three layers are mostly within 10 ppbv, with small low biases over the northern mid-latitudes (-6.2~-0.8 ppbv), and high biases

over the tropics in the lower and middle troposphere (e.g., about 10 ppbv at 800 hPa over the SH tropics), consistent with the comparison with satellite observations (Fig.4). We find that the model has large low ozone bias in the upper troposphere (300 hPa) particularly over the northern polar regions (~-30 ppbv). The underestimation extends to the stratosphere globally except for the extratropical Southern Hemisphere (Fig. 5). These negative model biases are likely due to the use of a simplified stratospheric ozone scheme and/or errors in modeling dynamics of ozone exchange

between the stratosphere and the troposphere as will be discussed later, or the low model vertical resolution (26 layers).

Figure 7 compares the simulated ground-level ozone with more than 300 rural/remote sites (defined by a number of metrics including population density and nighttime lights data, Schultz et al., 2017b) around the world from the TOAR database. We average all observations within the same model grid square for statistical analyses. BCC-GEOS-Chem

v1.0 captures the spatial and seasonal distributions of global ground-level ozone with $r$ ranging from 0.53 to 0.59 (N=154). The annual mean model biases are 4.9 ppbv (15%) for all observations, with larger high bias in June-July-August period (11.0 ppbv, 32%). Inclusion of urban and suburban sites slightly decreases the spatial correlations ($r$=0.34~0.60, N=292) and enlarges the annual mean high bias (10.2 ppbv). We find again that the high biases are more prominent in the tropics (e.g., coastal sites in the western Pacific and Indonesia) in summer. Although the above

comparison is heavily weighted toward the US, Europe, Japan and South Korea due to the density of observations in these regions, our results demonstrate the overall good performance for BCC-GEOS-Chem v1.0 in simulating ground-level ozone at least for rural and remote regions.

### 3.3 Tropospheric ozone and OH budgets in BCC-GEOS-Chem v1.0

We then diagnose the global tropospheric ozone burden and its driving terms (Table 4 and Figure 8). BCC-GEOS-Chem

v1.0 estimates the global tropospheric ozone burden to be 336.0 Tg averaged over 2012-2014. This is consistent with the results from the classic offline GEOS-Chem CTM and the G5NR-Chem (~350 Tg) with an earlier version (v10-01) of GEOS-Chem as chemical module (Hu et al., 2017; 2018), and also in agreement with the recent model assessments of 49 models (320-370 Tg, Young et al., 2018). We divide the global tropospheric ozone burden into different regions following Young et al. (2013) as shown in Figure 8a and 8b. We find that the overall distributions of ozone burden are

consistent with the ensemble of 15 models from the Atmospheric Chemistry and Climate Model Intercomparison Project (ACCMIP) (Young et al., 2013). The main discrepancy between BCC-GEOS-Chem v1.0 and ACCIMIP occurs within



30S-30°N. ACCMIP results show that ozone over 30°S-30°N and below 250 hPa accounts for 36.9% of the global tropospheric ozone burden, while BCC-GEOS-Chem v1.0 shows a higher proportion (48.5%). While ozone overestimation of BCC-GEOS-Chem v1.0 over 30°S-30°N is also seen from the comparisons to observations as

discussed previously, the discrepancy between our results and ACCMIP model ensemble mean is also likely due to the different simulation year (2000 conditions for ACCMIP versus 2012-2014 for BCC-GEOS-Chem v1.0). Zhang et al. (2016) showed that the equatorward redistribution of anthropogenic emissions significantly increased the global tropospheric ozone burden from 1980 to 2010, with the largest enhancements over the tropics. BCC-GEOS-Chem v1.0 underestimates the proportion of ozone burden in the upper troposphere (5.1%-10.9%) compared to the ACCMIP results

(6.4-15.2%), again likely reflecting the model limitation in simulating stratosphere ozone and/or its exchange with the troposphere.

We find that the global tropospheric mean OH concentration in BCC-GEOS-Chem v1.0 is $1.16 \times 10^6$ molecule cm$^{-3}$, close to the offline GEOS-Chem v10-01 ($1.25 \times 10^6$ molecule cm$^{-3}$, Hu et al., 2018) and well within the range of 16

ACCMIP models ($1.11 \pm 0.16 \times 10^6$ molecule cm$^{-3}$, Naik et al., 2013). Figure 8c and 8d compares the distribution of simulated OH concentrations with the climatology derived from previous studies (Spivakovsky et al., 2001; Emmons et al., 2010). We find that the model shows notable high bias in the lower troposphere (below 750 hPa) particularly in the tropics (2.04 to 2.45 molecule cm$^{-3}$ in BCC-GEOS-Chem v1.0 compared to 1.44 to 1.52 molecule cm$^{-3}$ in Spivakovsky et al., 2001). This discrepancy appears to be mainly driven by the high bias in ozone levels in this region. Furthermore,

we calculate the methane chemical lifetime against OH loss to be 8.3 years in BCC-GEOS-Chem v1.0, which falls in the low end of the range reported from ACCMIP multi-model assessments ($9.7 \pm 1.5$ years) (Naik et al., 2013).

We now diagnose the budget of global tropospheric ozone in BCC-GEOS-Chem v1.0. Following the classic GEOS-Chem, BCC-GEOS-Chem v1.0 diagnoses the chemical production and loss of the odd oxygen family ($O_x$, including $O_3$,

$NO_2$, $NO_y$, several organic nitrates and bromine species) to account for the rapid cycling among $O_x$ constituents. Ozone accounts for more than 95% of the total $O_x$ (Hu et al., 2017). The global annual ozone chemical production and loss are 5486 Tg and 4983 Tg, respectively (Table 4); both are higher than the classic GEOS-Chem (Hu et al., 2017) and fall in the high quartile of multi-model assessments (Young et al., 2018). The high tropospheric ozone production is due at least in part to the high precursor emissions used in this study particularly for $NO_x$ emissions The model shows strong

chemical production over northern mid-latitude continents in summertime, and large chemical loss over the tropical oceans driven by high water vapor content (figure not shown).

The global annual mean ozone dry deposition flux diagnosed in BCC-GEOS-Chem v1.0 is 873 Tg averaged for 2012-



2014. It is consistent with recent reviews by Hardacre et al. (2015) and Young et al. (2018) (700-1500 Tg from 33 model

estimates). Figure 9 presents the global ozone dry deposition velocity and flux for January and July 2012-2014. Both hemispheres show larger ozone dry deposition velocities in summer than winter due to stronger atmospheric turbulence and larger vegetation cover. Large ozone dry deposition velocity ($> 0.5$ cm s$^{-1}$) can be seen over the tropical continents, while over the oceans and glaciers ozone dry deposition is very weak.

We then diagnose the annual amount of ozone stratosphere-troposphere exchange (STE) of 370 Tg as the residual of mass balance between tropospheric chemical production, chemical loss, and deposition as previous studies did (Lamarque et al., 2012; Hu et al., 2017). This value is lower than most of other model estimates (400-680 Tg, Young et al., 2018). The low STE in BCC-GEOS-Chem v1.0 appears to be the main factor causing ozone underestimates in the upper troposphere as seen above. This may reflect a number of model limitations, for example, the representation of

stratospheric chemistry, inadequate STE due to model meteorology (e.g., biases in wind and tropopause), and the low model vertical resolution. Given the tropospheric ozone burden and its loss to chemistry and deposition, we derive the lifetime of tropospheric ozone of 20.9 days, consistent with the multi-model estimates (Young et al., 2013).

### 3.4 Evaluation of other atmospheric constituents

Figure 10 compares the model simulated spatial distributions of annual mean simulated NO$_2$, SO$_2$, CO, and CH$_2$O with

satellite observations. Here we do not apply the averaging kernels to smooth modeled results, and therefore mainly focus the comparisons on spatial variations rather than absolute magnitudes. As shown in Figure 10, BCC-GEOS-Chem v1.0 captures the observed hotspots of tropospheric NO$_2$ and PBL SO$_2$ columns over the East Asia that generally follow the distribution of anthropogenic sources. The sharp land-ocean gradients for both tracers reflect their short chemical lifetime. The spatial correlations between observations and model results are 0.87 for NO$_2$ and 0.52 for SO$_2$. The model

reproduces the large total CO column over the northern mid-latitudes driven by higher anthropogenic sources, and over the central Africa driven by biomass burning emissions ($r$=0.95). High levels of tropospheric CH$_2$O column are simulated over the Amazon, the central Africa, tropical Asia, and the southeastern US, typical regions where CH$_2$O oxidized from large biogenic emissions of VOCs ($r$=0.67). More assessments are required to correct the biases of these gaseous pollutants.


We evaluate model simulated AOD at 550 nm with the MODIS AOD observations in Figure 11. High AOD values over the East Asia due to high anthropogenic emissions, and over Africa and the adjacent oceans due to dust emissions are shown in both MODIS observations and BCC-GEOS-Chem v1.0, although the model tends to underestimate the observed hotspots likely due to the coarse model resolution. Figure 12 further shows the comparison of simulated surface



410 aerosol components (sulfate, nitrate, OC and BC) with the observations from the IMPROVE network over the US. The model fairly reproduces the spatial and seasonal patterns for all analyzed aerosol components, e.g., high sulfate and nitrate concentrations over the eastern US. Among all the components, the simulation of sulfate in the US shows best agreement with biases of -10%~20% and spatial correlation coefficients of 0.76-0.87 over model grids covering the measurement sites (N=77). The model also captures the high summertime OC and BC concentrations in the mid-western

415 US driven by active wildfire activities ($r$=0.20-0.57 for different seasons). However, the model shows high biases in wintertime nitrate in the eastern US as found in previous GEOS-Chem evaluations (Zhang et al., 2012).

## 4. Summary and future plans

This study describes the framework and evaluation of the new global atmospheric general circulation-chemistry model BCC-GEOS-Chem v1.0. The development of the BCC-GEOS-Chem v1.0 takes advantage of grid-independent structure

420 of the GEOS-Chem chemical module, which allows the exact same GEOS-Chem chemistry and deposition algorithms to be performed on any external grid and supported by MPI. BCC-GEOS-Chem v1.0 includes interactive atmospheric and land modules. It simulates the evolution of atmospheric chemical interactive constituents through a detailed mechanism of $HO_x$-$NO_x$-VOCs-ozone-bromine-aerosol tropospheric chemistry as well as online wet and dry deposition schemes. The model also implements a number of climate-sensitive natural emissions such as biogenic VOCs and

425 lighting NO.

We conduct a three-year (2012-2014) model simulation with year-specific CMIP6 anthropogenic and biomass burning emissions. We evaluate the model with a focus on tropospheric ozone using surface, ozonesonde, and satellite observations. We show that BCC-GEOS-Chem v1.0 can well capture the spatial distributions ($r$=0.79~0.93 with OMI

430 satellite observations of ozone at 700-400hPa) and seasonal cycles of tropospheric ozone. The model shows no significant biases in the lower and middle tropospheric ozone compared to satellite observations (0.4~2.2 ppbv at 700-400 hPa), ozonesonde (within 10 ppbv at 800, 500, and 300 hPa except for the polar upper troposphere), and surface measurements (4.9 ppbv). We calculate a global tropospheric ozone burden of 336 Tg year$^{-1}$ and OH burden of $1.16\times10^6$ molecule cm$^{-3}$; both are well within the ranges reported by previous studies. Regionally, the model shows notable high

435 biases in ozone over the tropics and low ozone biases in the upper troposphere. Model diagnostics show that BCC-GEOS-Chem v1.0 has higher tropospheric ozone chemical production and loss compared to the classic GEOS-Chem but still falls in the range of previous estimates. Comparisons of other air pollutants including $NO_2$, $SO_2$, CO, $CH_2O$, and aerosols show reasonable agreements, with biases likely due to uncertainties in emissions.

440 The development of BCC-GEOS-Chem v1.0 for online atmospheric chemistry simulation represents an important step





for the development of fully-coupled earth system models in China. There are still several limitations in this version that should be addressed in future model development. The current version of BCC-GEOS-Chem does not include full stratospheric chemistry mechanism, which is important for accurately modeling the evolution of ozone and its climate influences (Lu et al., 2019). We plan to implement the unified tropospheric-stratospheric chemistry extension (UCX)

(Eastham et al., 2014), which is now the "Standard" mechanism for GEOS-Chem chemistry, into the next version of BCC-GEOS-Chem. Diagnosing radiative transfer and aerosol-cloud interactions will be the next priority for model evaluation, and it can take advantage of the GEOS-Chem aerosol microphysics module (TwO-Moment Aerosol Sectional (TOMAS) module (Kodro and Pierce, 2017) or Advanced Particle Microphysics (APM) (Yu and Luo, 2009)). Updates of emissions (e.g., application of new or regional anthropogenic emissions inventories) could be merged to

BCC-GEOS-Chem with the future implementation of the GEOS-Chem emission module (Harvard-NASA Emissions Component, HEMCO) (Keller et al., 2014). BCC-GEOS-Chem is ready to be updated to higher horizontal and vertical resolution of T106 (about 110km, 46 layers up to 1.5hPa) or T266 (about 45 km, 56 layers up to 0.09 hPa) with recent BCC-CSM-MR and BCC-CSM-HR (Wu et al., 2019), which enables applications on air quality prediction in the future.

**Code and data availability**

The GEOS-Chem model is maintained at the Harvard Atmospheric Chemistry Modeling group (http://acmg.seas.harvard.edu/geos/). The source code of BCC-GEOS-Chem v1.0 can be accessed at a DOI repository https://doi.org/10.5281/zenodo.3475649, and model outputs for 2012-2014 are available at https://doi.org/10.5281/zenodo.3496777. All source code and data can also be accessed by contacting the corresponding

authors Lin Zhang (zhanglg@pku.edu.cn) and Tongwen Wu (twwu@cma.gov.cn).

**Author Contributions**

Lin Zhang, Tongwen Wu, Daniel Jacob, and Jun Wang led the project. Xiao Lu, Lin Zhang, Tongwen Wu, Michael Long, Fang Zhang and Jie Zhang developed the model source code. Xiao Lu performed model simulations, analyzed

data, and prepared the figures with suggestions from all authors. Xiao Lu, Lin Zhang, Tongwen Wu, Daniel Jacob wrote the paper. All authors contributed to the discussion and improvement of the paper.

**Competing interests**

The authors declare that they have no conflict of interest.


**Acknowledgement**

This work is supported by the National Key Research and Development Program of China (2017YFC0210102) and the



Harvard China Project. Xiao Lu is also supported by the Chinese Scholarship Council.

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





**Table 1. Mapping of land use types (LUT) used in BCC-GEOS-Chem v1.0 to the Wesely deposition surfaces for deposition. Also shown are the roughness heights ($Z_o$) for each LUT.**

|    | BCC-AVIM LUT | GEOS-Chem LUT for dry deposition | $Z_o$ (m) |
|----|--------------|----------------------------------|-----------|
| 0  | bareground | desert | 0.001 |
| 1  | needleleaf evergreen temperate tree | coniferous forest | 1 |
| 2  | needleleaf evergreen boreal tree | coniferous forest | 1 |
| 3  | needleleaf deciduous boreal tree | coniferous forest | 1 |
| 4  | broadleaf evergreen tropical tree | Amazon forest | 1 |
| 5  | broadleaf evergreen temperate tree | deciduous forest | 1 |
| 6  | broadleaf deciduous tropical tree | deciduous forest | 1 |
| 7  | broadleaf deciduous temperate tree | deciduous forest | 1 |
| 8  | broadleaf deciduous boreal tree | deciduous forest | 1 |
| 9  | broadleaf evergreen shrub | shrub/grassland | 0.01 |
| 10 | broadleaf deciduous temperate shrub | shrub/grassland | 0.01 |
| 11 | broadleaf deciduous boreal shrub | shrub/grassland | 0.01 |
| 12 | C3 arctic grass | tundra | 0.002 |
| 13 | C3 non-arctic grass | tundra | 0.01 |
| 14 | C4 grass | tundra | 0.01 |
| 15 | crop | agricultural land | 0.1 |
| 16 | wheat | agricultural land | 0.1 |
| 17 | ocean | water | 0.001 |
| 18 | glacier | snow/ice | 0.0001 |
| 19 | lake | water | 0.001 |
| 20 | wetland | wetland | 0.05 |
| 21 | urban | urban | 2.5 |






**Table 2. Global annual emissions used in the BCC-GEOS-Chem v1.0 categorized by sectors in unit of Tg year$^{-1}$**

| Species | Anthropogenic | Biomass burning | Biogenic | Ocean | Soil | Aircraft | Others | Total |
|---|---|---|---|---|---|---|---|---|
| NO | 91.5 | 6.4 | | | 11.2 | 2.0 | Lightning: 11.5 | 122.6 |
| CO | 617.2 | 231.8 | 75.9 | | | 0.6 | | 925.5 |
| ALK4($C_4H_{10}$) | 17.7 | 0.2 | 26.1 | | | | | 44.0 |
| ALK5($C_5H_{12}$) | 21.4 | | | | | | | 21.4 |
| ALK6($C_6H_{14}$) | 26.5 | | | | | | | 26.5 |
| Acetone ($CH_3COCH_3$) | 1.1 | 3.1 | 19.2 | 9.9 | | | | 33.3 |
| ALD2($CH_3CHO$) | 1.2 | 2.4 | | | | | | 3.6 |
| ISOP | | | 410.0 | | | | | 410.0 |
| $C_2H_4$ | 5.9 | 3.2 | 7.5 | | | | | 16.6 |
| $C_3H_6$ (PRPE) | 3.6 | 2.5 | 10.7 | | | | | 16.8 |
| $C_3H_8$ | 6.7 | 0.5 | | | | | | 7.2 |
| $CH_2O$ | 2.5 | 3.2 | | | | | | 5.7 |
| $C_2H_6$ | 6.6 | 2.7 | | | | | | 9.3 |
| BENZ($C_6H_6$) | 6.7 | | | | | | | 6.7 |
| TOLU($C_7H_8$) | 7.8 | | | | | | | 7.8 |
| XYLE($C_8H_{10}$) | 7.5 | | | | | | | 7.5 |
| $SO_2$ | 112.5 | 1.7 | | | | 0.3 | Volcano: 9.2 | 123.7 |
| $NH_3$ | 60.1 | 2.9 | | 8.2 | 2.4 | | | 73.6 |
| DMS | | | | 27.4 | | | | 27.6 |
| BC | 7.9 | 1.3 | | | | <0.1 | | 9.2 |
| OC | 19.5 | 11.5 | | | | <0.1 | | 31.0 |



**Table 3. Observational datasets used for model evaluation**

| Species | Observation | Horizontal Resolution | Vertical Levels | Data sources or reference |
|---|---|---|---|---|
| Ozone | WOUDC network | | Vertical profile | http://woudc.org/data.php (last access: 2 June 2019) |
| | TOAR dataset | | Surface | Schultz et al. (2017a, https://doi.org/10.1594/PANGAEA.876108, last access: 2 June 2019) |
| | OMI satellite | 2°×2.5° | 24 layers | Liu et al. (2010) |
| CO | MOPITT satellite | 1°×1° | Total column | https://www2.acom.ucar.edu/mopitt |
| $NO_2$ | OMI satellite | 0.25°×0.25° | Tropospheric column | https://disc.gsfc.nasa.gov/datasets/OMNO2d_003/summary (last access: 2 June 2019) (Krotkov, 2013) |
| $CH_2O$ | OMI satellite | 0.25°×0.25° | Tropospheric column | http://h2co.aeronomie.be/ (last access: 2 June 2019) (De Smedt et al., 2015) |
| $SO_2$ | OMI satellite | 0.25°×0.25° | Tropospheric column | https://disc.gsfc.nasa.gov/datasets/OMSO2e_003/summary (last access: 2 June 2019) (Krotkov et al., 2015) |
| AOD | MODIS | 1°×1° | Atmosphere | (https://neo.sci.gsfc.nasa.gov/view.php?datasetId=MODAL2_M_AER_OD, last access: 2 June 2019) |
| Aerosol composition | IMPROVE network | | Surface | Interagency Monitoring of Protected Visual Environments (IMPROVE) (http://vista.cira.colostate.edu/Improve/, last access: 7 July 2019) |





**Table 4. Global budget of tropospheric ozone diagnosed in BCC-GEOS-Chem v1.0 and comparison with other studies.**

| Diagnostic term | BCC-GEOS-Chem (this study) | Classic GEOS-Chem (Hu et al., 2017)[a] | Other references |
|---|---|---|---|
| Ozone burden (Tg) | 336 | 351 | mean: 340, range: 250-410[b] |
| $O_x$ chemical production (Tg year$^{-1}$)[c] | 5486 | 4960 | mean: 4900, range: 3800-6900[d] |
| $O_x$ chemical loss (Tg year$^{-1}$) | 4983 | 4360 | mean: 4600, range: 3300-6600[e] |
| Dry deposition (Tg year$^{-1}$) | 873 | 908 | mean: 1000, range: 700-1500[f] |
| STE (Tg year$^{-1}$) | 370[g] | 325[g] | mean: 500, range: 180-920[h] |
| Lifetime (days) | 20.9 | 24.2 | mean: 22.3, range:19.9-25.5[i] |
| Global OH ($10^6$ molecule cm$^{-3}$) [j] | 1.16 | 1.25 | mean±STD: 1.11±0.16, range: 0.74-1.33[k] |
| Methane chemical lifetime (years) | 8.27 | | mean±STD: 9.7±1.5, range: 7.1-14.0[k] |

[a] from Table 2 in Hu et al. (2017). The GEOS-Chem version is v10-01.

[b] from Figure 3 in Young et al. (2018), 49 models for 2000 condition.

[c] Budget is for the odd oxygen family, including $O_3$, $NO_2$, $NO_y$, several organic nitrates and bromine species) to account for the rapid cycling among $O_x$ constituents. Ozone accounts for more than 95% of the total $O_x$.

[d] from Figure 3 in Young et al. (2018), 33 models for 2000 condition.

[e] from Figure 3 in Young et al. (2018), 32 models for 2000 condition.

[f] from Figure 3 in Young et al. (2018), 33 models for 2000 condition.

[g] estimated from the residual of mass balance between tropospheric chemical production, chemical loss, and deposition.

[h] from Figure 3 in Young et al. (2018), 34 models for 2000 condition.

[i] from Table 2 in Young et al. (2013), 6 models for 2000 condition.

[j] Global annual mean air-mass-weighted OH concentration in the troposphere

[k] from Table 1 in Naik et al. (2013), 16 models for year 2000. STD stands for standard deviation.



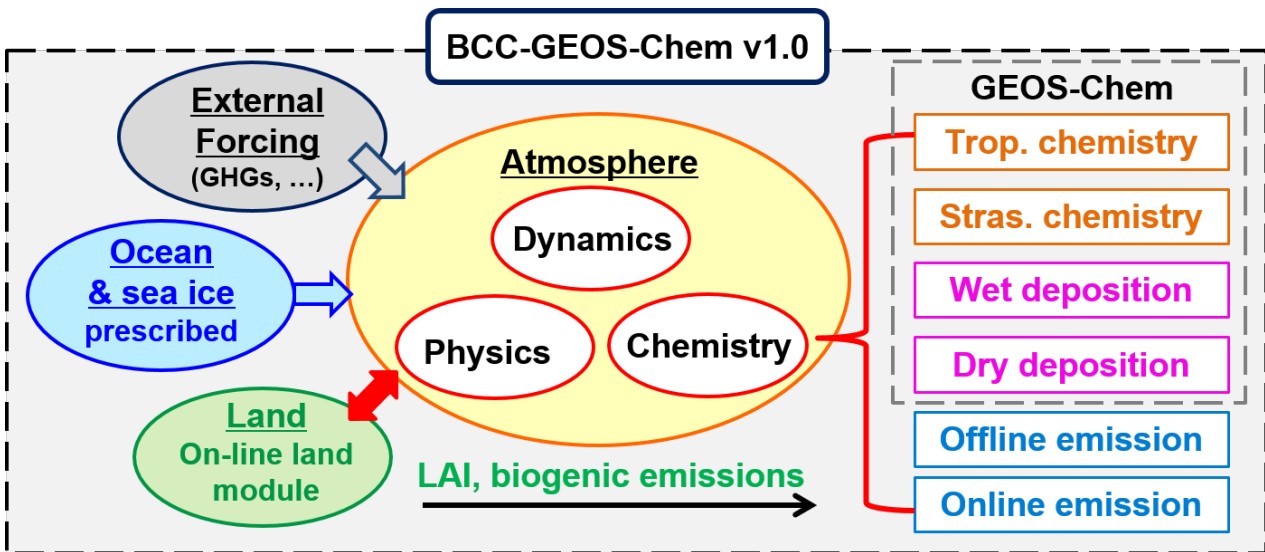

**Figure 1.** Schematic diagram of the BCC-GEOS-Chem v1.0 model framework.



**Figure 2.** Spatial distributions of annual total emissions used in the study, (a) total NO emission (not including lightning emissions); (b) total CO emission; (c) biogenic isoprene emission; (d) lightning NO emission; (e) sea salt emission (dry mass); 800 (f) mineral dust emissions.



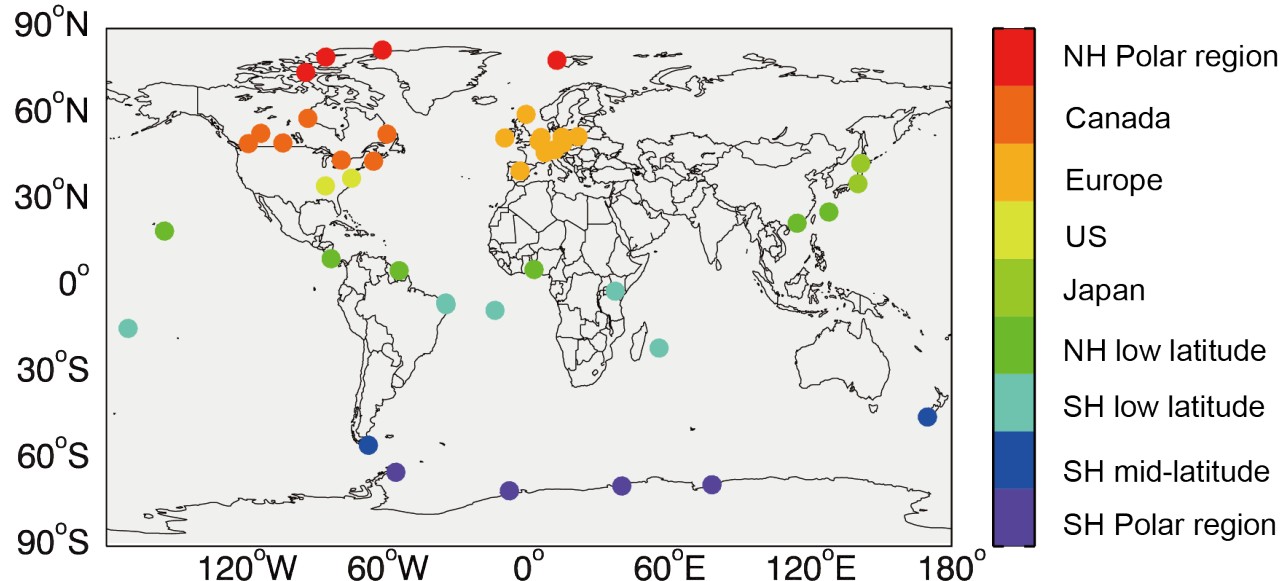

**Figure 3.** Locations of selected ozonesonde observations in 2012-2014 used in this study categorized by nine regions.




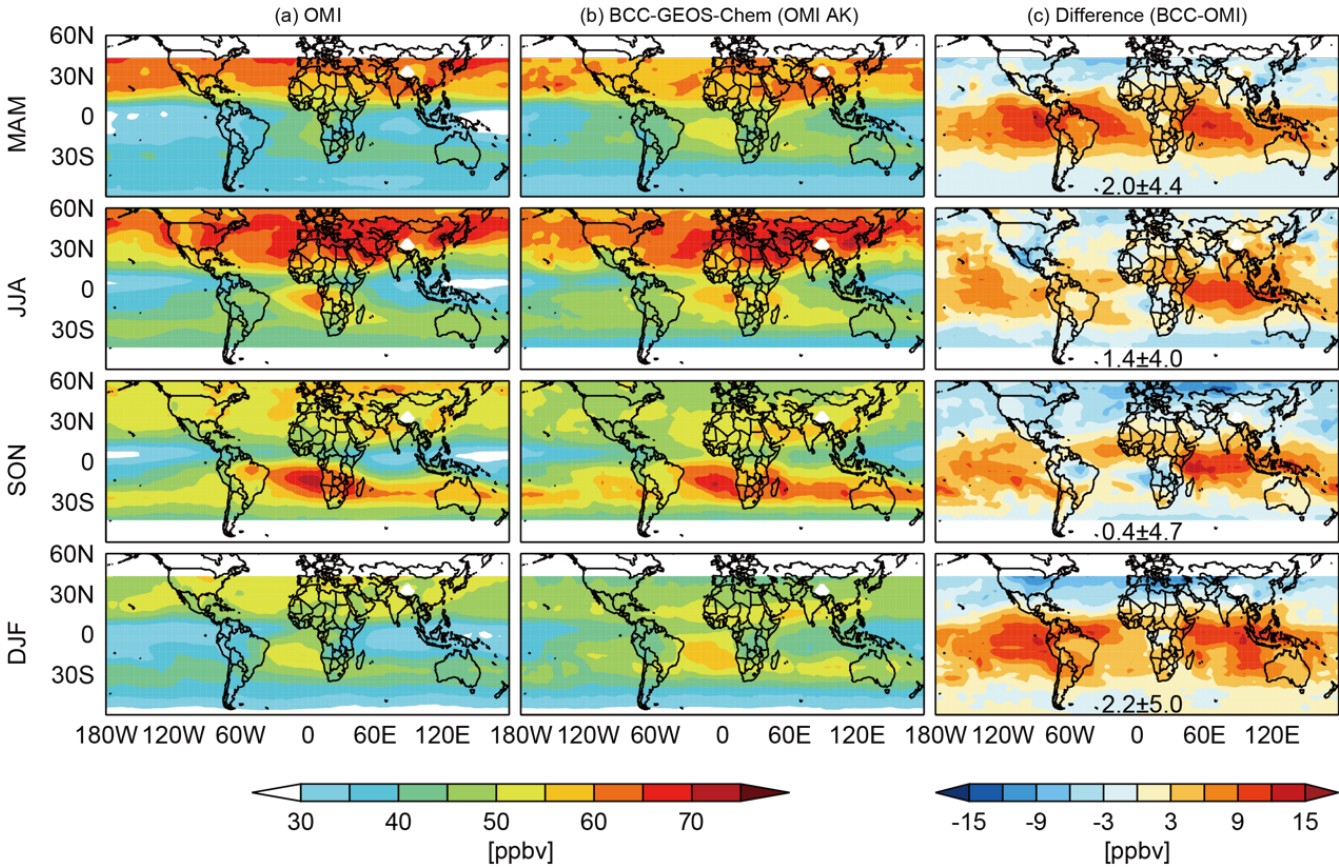

**Figure 4.** Spatial and seasonal distributions of tropospheric ozone at 700-400 hPa from (a) OMI satellite observations; (b) BCC-GEOS-Chem v1.0 model results (with OMI averaging kernels applied), and (c) differences between the two (model results minus observations) with the seasonal mean differences (±standard deviations) shown inset. Values are 3-year averages for 2012-2014.





**Figure 5.** Comparisons of BCC-GEOS-Chem v1.0 simulated ozone vertical profiles to ozonesonde observations averaged over the nine regions (Fig. 3) from south to north. Black horizontal bars are the standard deviations of observations. Numbers of available sites and records for each region in 2012-2014 are given inset.

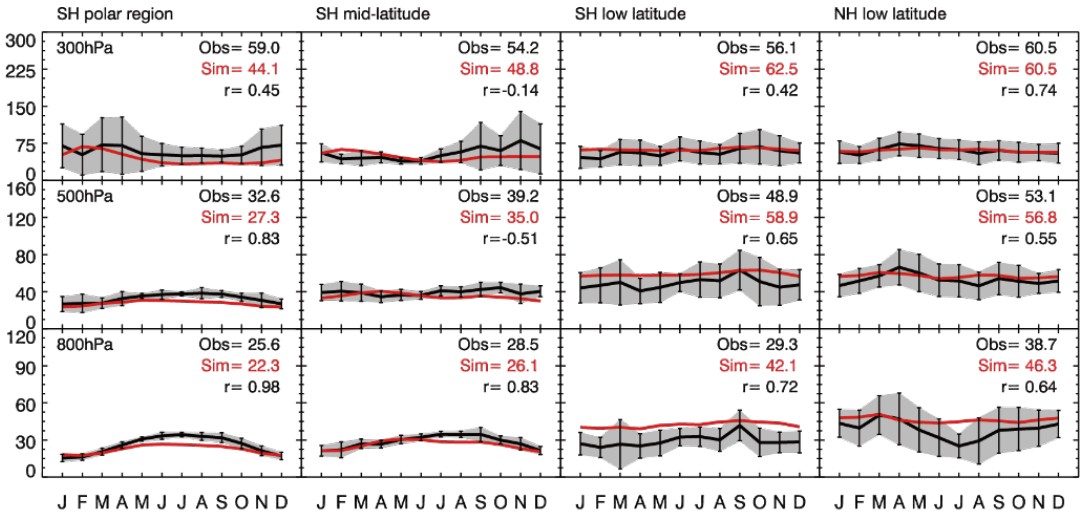

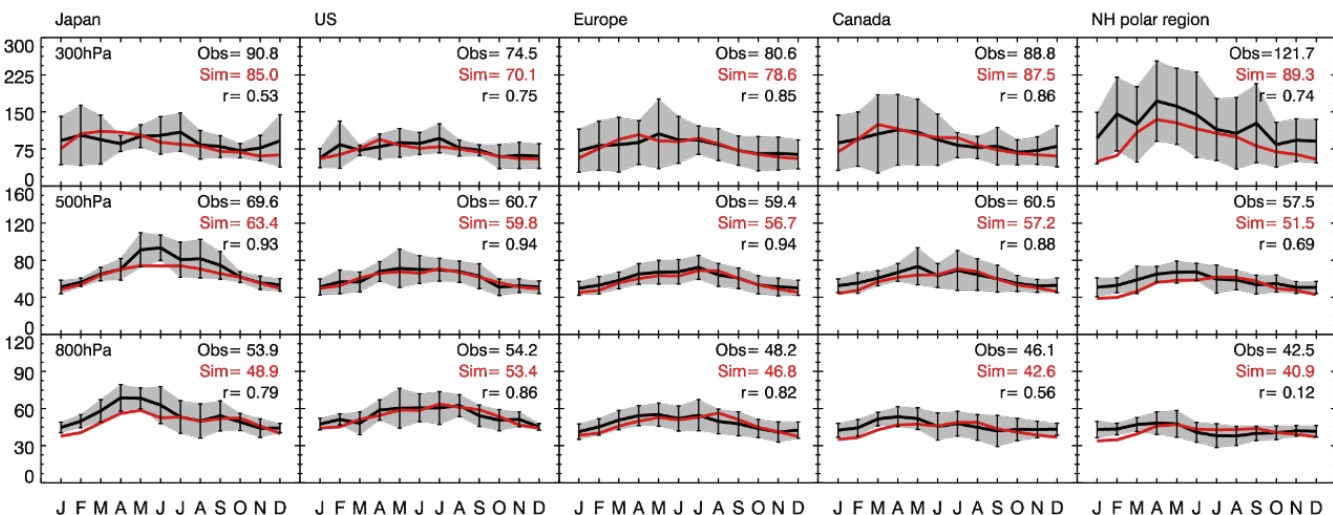

**Figure 6.** Seasonal variations of ozonesonde observed (black lines) and model simulated ozone (red lines) at three pressure levels (300 hPa, 500 hPa, and 800 hPa) averaged over the nine regions (Fig.3). Vertical black bars and grey shadings are the standard deviations of observations. The annual means of observed and simulated ozone concentrations and their correlation coefficients are shown inset. Values are 3-year averages for 2012-2014.


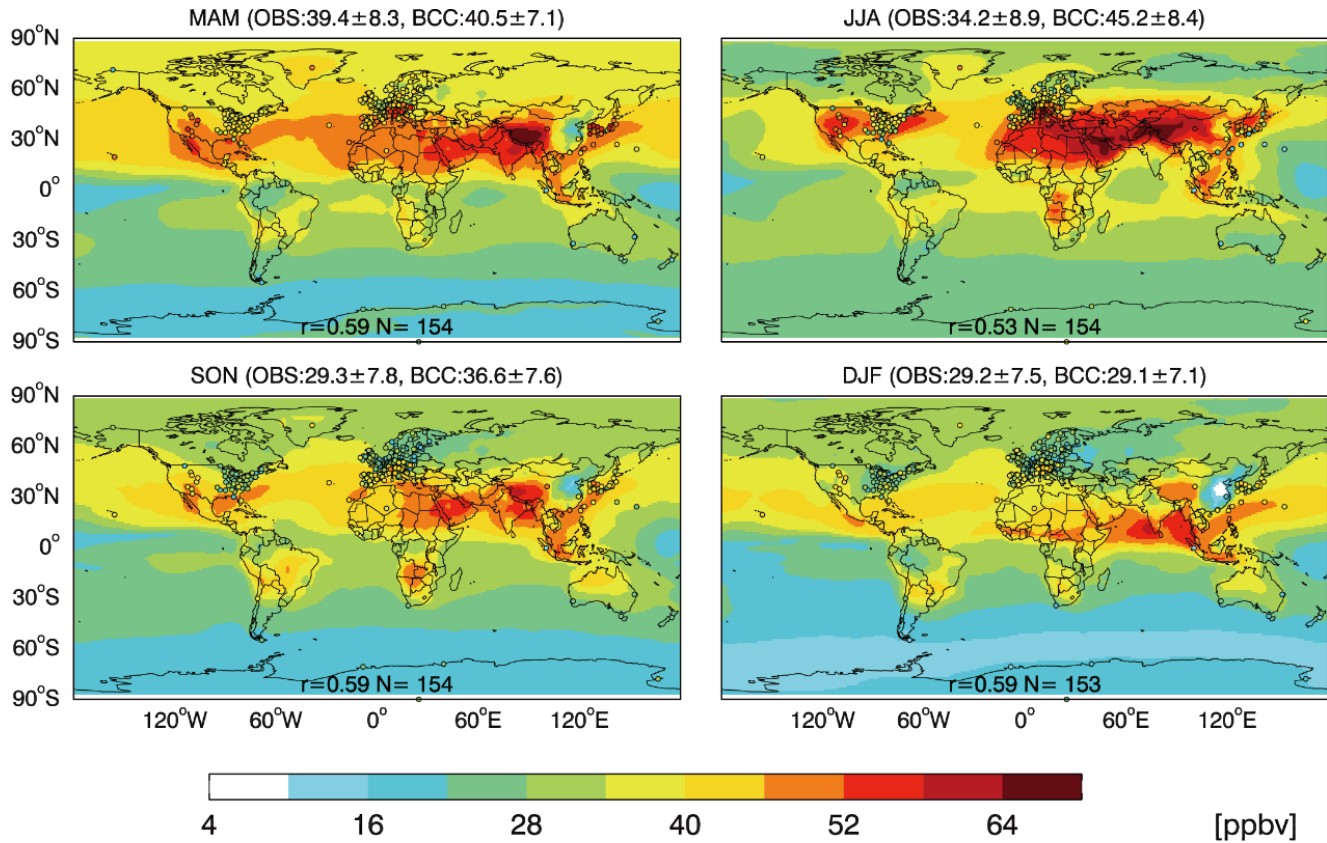

**Figure 7.** Spatial and seasonal distributions of simulated surface ozone mixing ratios (contours) over 2012-2014. The model results are compared to observations at rural/remote sites (circles) from the TOAR dataset. Seasonal mean values for observations and model results (sampled at corresponding TOAR data sites), their spatial correlation coefficients, and the number of co-sampled grids are shown inset.



**Figure 8.** Zonal and vertical distributions of the tropospheric ozone burden and OH concentrations. For comparison, panel (a)
shows tropospheric ozone burden in the year 2000 from Young et al. (2013) based on 15 ACCMIP models, and panel (c)
shows climatological tropospheric OH burden reported by Spivakovsky et al. (2000) and summarized by Emmons et al. (2010).
Panel (b) and (d) show corresponding results from BCC-GEOS-Chem v1.0 averaged over 2012-2014. The red lines in (a) and
(c) denote the tropopause derived from the National Centers for Environmental Prediction (NCEP) reanalysis, and these in (b)
and (d) are from BCC-GEOS-Chem v1.0.


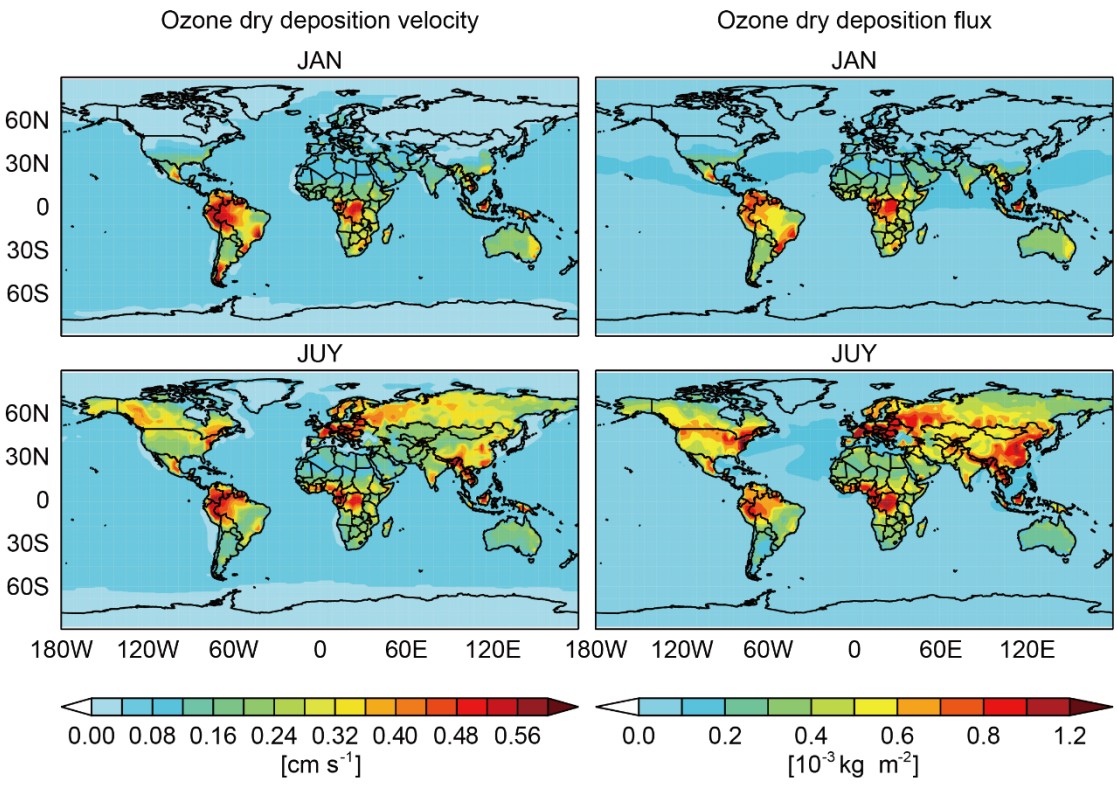

**Figure 9.** Monthly mean model diagnosed ozone deposition velocities (cm s$^{-1}$) and fluxes (kg m$^{-2}$) in January (top panels) and July (bottom panels).






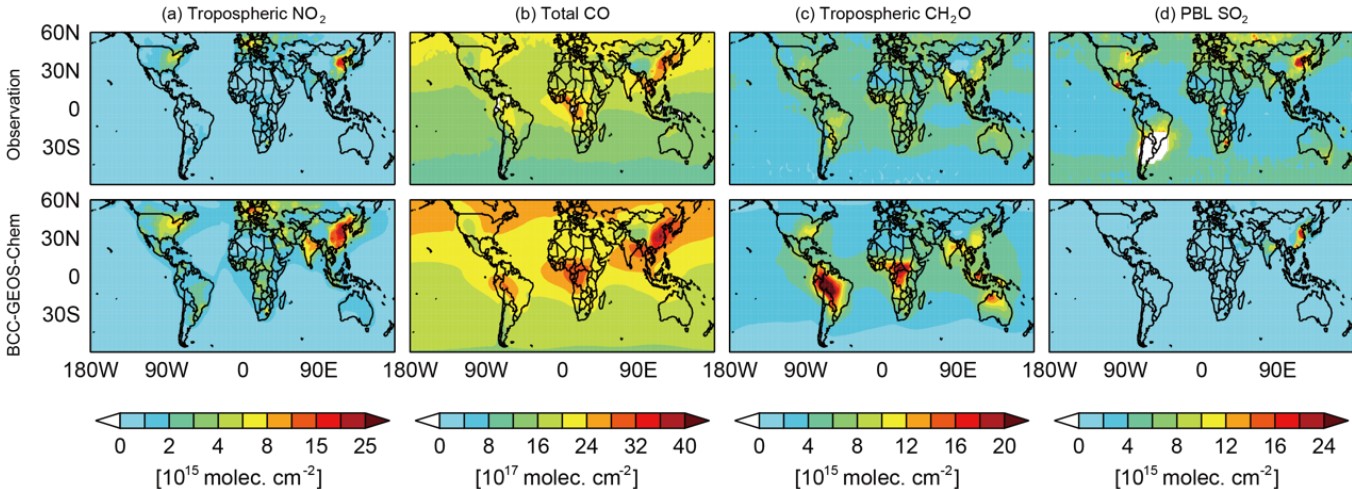

**Figure 10.** Spatial distributions of satellite observed (top panels) and model simulated (bottom panels) annual mean (a) tropospheric $NO_2$ column, (b) total CO column, (c) tropospheric $CH_2O$ column, (d) $SO_2$ column in planetary boundary layer. Values are 3-year averages for 2012-2014.

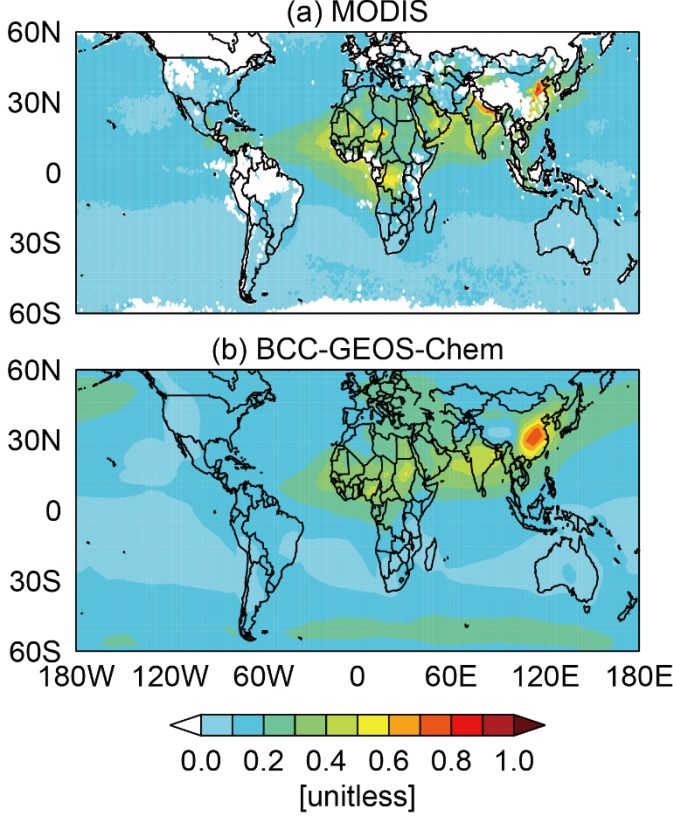

**Figure 11.** Spatial distributions of annual mean aerosol optical depth (AOD) at 550 nm from (a) MODIS satellite observations and (b) BCC-GEOS-Chem v1.0 averaged over 2012-2014.



**Figure 12.** Spatial and seasonal distributions of simulated surface concentrations (contours) of (a) aerosol sulfate, (b) nitrate, (c) organic carbon, and (d) black carbon compared with observations from the US IMPROVE network (circles) over 2012-
2014. Correlation coefficients between observations and model results sampled at the site locations are shown inset.