# Peer review of "Development of the global atmospheric general circulation-chemistry model BCC-GEOS-Chem v1.0: model description and evaluation"

_Geoscientific Model Development, 2019_

## Referee Comment (RC1) · Anonymous Referee #1 · 4 Mar 2020

This paper describes a framework and evaluation of a novel model framework which incorporates the GEOS-Chem 1-D atmospheric chemistry component into a the Beijing Climate Center's Atmospheric GCM (BCC-AGCM). With this combination successfully established, the authors evaluate the model in comparison with a suite of observations, including for tropospheric ozone, OH concentrations, and methane chemical lifetimes. They also compare satellite observations of various important atmospheric measurements, including $NO_2$, CO, $SO_2$, $CH_2O$, and AOD. The work is a significant step forward in the development of Earth System Models in China and is clearly relevant to readers of GMD, and should be accepted pending the authors addressing a few relatively minor points relating to the reproducibility and presentation quality of their

manuscript.

Specific Comments: Table 3: it is not clear how oceanic trace gas emissions are parameterized in this work. The work does reference the use of CMIP5 DMS emissions from the ocean in line 204, but Table 3 shows that an oceanic emission of acetone was also included. While this inclusion makes sense given recent research on the subject, it is not clear how this source was decided upon or how it was represented. The representation of the oceanic source of NH3 listed in Table 3 is also unexplained. Additionally, the ocean is a known source of acetaldehyde (Millet et al., 2010; Wang et al., 2019), but this source is not accounted for in Table 3. It is not discussed in the paper why this is the case.

References: Millet, D. B., Guenther, A., Siegel, D. A., Nelson, N. B., Singh, H. B., de Gouw, J. A., et al. (2010). Global atmospheric budget of acetaldehyde: 3‐D model analysis and constraints from in‐situ and satellite observations. Atmospheric Chemistry and Physics, 10(7), 3405–3425. https://doi.org/10.5194/acp-10-3405-2010

Wang, S., Hornbrook, R. S., Hills, A., Emmons, L. K., Tilmes, S., Lamarque, J.‐F., et al. (2019). Atmospheric acetaldehyde: Importance of air‐sea exchange and a missing source in the remote troposphere. Geophysical Research Letters, 46(10), 5601–5613. https://doi.org/10.1029/2019GL082034

Figure 6: The y-Axis is unlabeled and not clearly explained in the caption. I assume this is ozone in ppb, but some label or caption edit is probably in order. More importantly, it is not clear from the discussion of Figure 6 and Figure 3 whether the simulated values are taken from GEOS-Chem over some defined region or simply from the grid boxes corresponding to the observations in Figure 3.

I have questions about the choices of regions used in Figures 3 and 6, though I recognize the authors are referencing regions already in the literature. In particular, I do not agree with the classification of the "NH Low Latitude" grouping in the context of this paper: lumping together southern China, Hawaii, Panama, French Guiana, and

Nigeria into one coherent region would to elide an enormous amount of difference in chemical regimes, biomass burning, and anthropogenic emissions between the various regions chosen. I believe it makes more sense to separate the points near China and Taiwan into a "Southeast Asia" region, separate from the remaining "NH Low Latitude" grouping. As the "SH midlatitude" region already has an N of 2, this should be similarly acceptable. This choice may have ramifications for the seasonal simulation performance as summarized in Figure 6.

Figure 7: The choice made to show observations as small circles on a global map is very confusing in this context. Given the density of spatial coverage over the Northeastern US and Europe, I suggest 3 possible fixes. 1) grid the observations to the same resolution as the model output and show them side-by-side with the simulated ozone; 2) show different plots for the US, for Europe, and for the rest of the globe if necessary; or 3) the points should be averaged for display purposes over a larger spatial extents and made larger themselves. The currently displayed global view could be placed in the SI. As of now, Figure 7 serves a mostly pro forma purpose – it is hard for me to glean any useful information from the plot given the display, and true patterns in the observation data (for example the increase in surface ozone in the observations over central Europe) are obscured by the chosen plotting scheme, rather than illustrated. If the authors do not believe that Figure 7 should be changed, perhaps it belongs in the SI in its entirety, as it is not clear to me what the figure adds to the paper in its current state.

Figure 8: This is an excellent figure – very clear, with a high information-to-ink ratio. One comment: it seems unlikely that the geographic equator is a meaningful division in this kind of global average, given that the 'meteorological equator' deviates from 0 degrees depending upon season and region. If the comparison datasets support the option, the authors should consider using 5 latitude bins instead of 4 – perhaps 90-50 S, 50-20 S, 20 S - 20 N, 20-50 N, and 50-90 N? This has the benefit of treating the tropics, midlatitudes, and polar-latitudes differently.

Figure 10: why is the discrepancy between modeled and observed SO2 not discussed further? It is true that the model does broadly reproduce the spatial trend observed over China and India but elsewhere the correlation would appear to be quite poor.

Technical Corrections: Line 448: Citation is misspelled – should be "Kodros and Pierce, 2017". The reference is correctly spelled in the bibliography.

---

## Referee Comment (RC2) · Anonymous Referee #2 · 8 Jun 2020

The manuscript presents an overview and assessment of a newly constructed chemistry-climate model that has resulted from the linking of GEOS-Chem with the Beijing Climate Center AGCM. The general features of the model are presented and a fairly extensive comparison against observations for ozone are presented. In addition, other aspects of the model chemical climate are presented, including the global distribution of OH, ozone budget terms, and some limited comparisons for aerosol quantities against observations including AOD and speciated aerosol concentrations over the US.

The manuscript is very clearly written and presented, providing a fairly complete overview of the model components and an idea of how the chemical climate of BCC-

[Figure]

GEOS-Chem compares with GEOS-Chem itself and with other chemistry climate models. My only significant criticism is focused on Figure 10, comparing column amounts of $NO_2$, CO, $SO_2$ and $CH_2O$. In the discussion of Figure 10, lines 394 – 404, the authors say that the averaging kernels for the satellite observations were not applied to the model concentrations when calculating the column amounts and that the comparisons 'mainly focus ... on spatial variations rather than absolute magnitudes.' In this case the comparison is nothing more than a test to make sure the specified emissions are being put into the model in the correct locations. Global models have a long-standing low bias for CO in the northern hemisphere that appears to be related to emissions and the hydroxy radical, but Figure 10 shows that BCC-GEOS-Chem has too high CO in the Northern Hemisphere. Due to the lack of a quantitative comparison with the satellite data by application of the averaging kernel it is impossible to judge whether the differences signify anything. While $CH_2O$ is not predominantly due to direct emission, the spatial distribution is tightly coupled to the emissions of biogenic hydrocarbons, so the comparison will also be largely driven by having regions of high biogenic emissions in the correct place. The differences in the magnitude of $CH_2O$ between the satellite and model is quite large and it would be interesting to have a more quantitative comparison with the satellite observations as the qualitative comparison focused on the spatial distribution is not informative at all. I would strongly urge the authors to revise the comparison of the column amounts to be more quantitative by application of the appropriate averaging kernel.

My other comments are all minor in nature and are given below.

Lines 80 - 83: Here the authors state 'Integration of GEOS-Chem chemical module into CSMs has been enabled by separating the module (which simulates all local processes including chemistry, deposition, and emission) from the simulation of transport, and making it operate on 1-D (vertical) columns in a grid-independent manner (Long et al., 2015; Eastham et al., 2018).' How is the 1-D column version of GEOS-Chem integrated with a 3-D CSM for processes that typically occur in the physics of the model such as

vertical turbulent diffusion and transport by deep convection? (I do find a description of deep convection and wet deposition around line 190, but no mention of how vertical diffusion is performed.)

Lines 181 – 183: The dry deposition uses the general characteristics of the land surface as given by the CSM land module BCC-AVIM. Are there also links to the land surface scheme for more short-term variables such as stomatal resistance, that would allow for effects such as drought on dry deposition?

Line 234: Minor typo in 'The model estimates t global annual ...'

Lines 311 – 314: Somewhere, either in the discussion of Figure 5 or the caption, there should be mention that the comparison is for annual average ozone.

Lines 314 – 321: I was a bit curious about why the vertical profile of ozone for the Japanese stations shows such a different vertical structure between the observations and model in Figure 5. Looking at Figure 6, the 300 hPa doesn't show that big of a difference. If 300 hPa is somewhere around 10 – 11 km, shouldn't the annual average in the observations be over 120 ppbv, though it is listed as 90 ppbv on Figure 6?

Line 364: Discussing the discrepancy in OH in the tropics between the Spivakovsky climatology the authors state 'This discrepancy appears to be mainly driven by the high bias in ozone levels in this region.' Attempts to understand the reasons for differences in OH between models has shown how many different factors play a role – see, for example, Nicely et al. Atmos. Chem. Phys. 20, doi:10.5194/acp-20-1341-2020, 2020. Do the authors have some reason to believe that the ozone and hydroxyl biases are related and, if not I would suggest removing this statement.

---

## Author Comment (AC1) · 10 Jul 2020

**Reply of RC1: Review of Lu et al., 2019 "Development of the global atmospheric general circulation-chemistry model BCC-GEOS-Chem v1.0: model description and evaluation"**

5  **Reviewer #1**
**Comment#1-1:** This paper describes a framework and evaluation of a novel model framework which incorporates the GEOS-Chem 1-D atmospheric chemistry component into the Beijing Climate Center's Atmospheric GCM (BCC-AGCM). With this combination successfully established, the authors evaluate the model in comparison
10  with a suite of observations, including for tropospheric ozone, OH concentrations, and methane chemical lifetimes. They also compare satellite observations of various important atmospheric measurements, including NO2, CO, SO2, CH2O, and AOD. The work is a significant step forward in the development of Earth System Models in China and is clearly relevant to readers of GMD, and should be accepted pending the authors
15  addressing a few relatively minor points relating to the reproducibility and presentation quality of their manuscript.
**Response#1-1: We thank the reviewer for the valuable comments. All of them have been implemented in the revised manuscript. Please see our itemized responses below.**

20

**Comment#1-2:** Specific Comments: Table 3: it is not clear how oceanic trace gas emissions are parameterized in this work. The work does reference the use of CMIP5 DMS emissions from the ocean in line 204, but Table 3 shows that an oceanic emission of acetone was also included. While this inclusion makes sense given recent research
25  on the subject, it is not clear how this source was decided upon or how it was represented. The representation of the oceanic source of NH3 listed in Table 3 is also unexplained. Additionally, the ocean is a known source of acetaldehyde (Millet et al., 2010; Wang et al., 2019), but this source is not accounted for in Table 3. It is not discussed in the paper why this is the case.

30

References:
Millet, D. B., Guenther, A., Siegel, D. A., Nelson, N. B., Singh, H. B., de Gouw, J. A., et al. (2010). Global atmospheric budget of acetaldehyde: 3-D model analysis and constraints from in-situ and satellite observations. Atmospheric Chemistry and Physics, 10(7), 3405–3425.
35  https://doi.org/10.5194/acp-10-3405-2010

Wang, S., Hornbrook, R. S., Hills, A., Emmons, L. K., Tilmes, S., Lamarque, J. F., et al. (2019). Atmospheric Acetaldehyde: Importance of Air-Sea Exchange and a Missing Source in the Remote Troposphere. Geophysical Research Letters, 46(10), 5601–5613.
40  https://doi.org/10.1029/2019GL082034
**Response#1-2: Thanks for pointing it out. The oceanic emissions of acetone and ammonia (NH3) are obtained from the Atmospheric Chemistry and Climate Model Intercomparison Project (ACCMIP) emission inventory (Lamarque et al., 2010). We indeed did not consider the oceanic acetaldehyde emissions, which**

45  **should be addressed in the next model version.**

**We now state in the Section 2.5.1 (Offline emissions): "We also incorporate
emissions from the Atmospheric Chemistry and Climate Model Intercomparison
Project (ACCMIP) inventory (http://accent.aero.jussieu.fr/ACCMIP.php, last**
50  **access: 14 Jun 2020; Lamarque et al., 2010) and from Wu et al. (2020) for emissions
not included in CEDS data set. These mainly apply to oceanic emissions, soil NO$_x$
emissions, and volcanic SO$_2$ emissions. Several sources (e.g., oceanic acetaldehyde
emissions (Millet et al., 2010; Wang et al., 2019)) have not yet been included in this
model version."**

55
**Reference added:**

Lamarque, J. F., Bond, T. C., Eyring, V., Granier, C., Heil, A., Klimont, Z., Lee, D., Liousse, C.,
    Mieville, A., Owen, B., Schultz, M. G., Shindell, D., Smith, S. J., Stehfest, E., Van Aardenne,
    J., Cooper, O. R., Kainuma, M., Mahowald, N., McConnell, J. R., Naik, V., Riahi, K., and van
60      Vuuren, D. P.: Historical (1850–2000) gridded anthropogenic and biomass burning emissions
    of reactive gases and aerosols: methodology and application, Atmos. Chem. Phys., 10, 7017-
    7039, http://doi.org/10.5194/acp-10-7017-2010, 2010.
Millet, D. B., Guenther, A., Siegel, D. A., Nelson, N. B., Singh, H. B., de Gouw, J. A., Warneke, C.,
    Williams, J., Eerdekens, G., Sinha, V., Karl, T., Flocke, F., Apel, E., Riemer, D. D., Palmer, P.
65      I., and Barkley, M.: Global atmospheric budget of acetaldehyde: 3-D model analysis and
    constraints from in-situ and satellite observations, Atmos. Chem. Phys., 10, 3405-3425,
    http://doi.org/10.5194/acp-10-3405-2010, 2010.
Wang, S., Hornbrook, R. S., Hills, A., Emmons, L. K., Tilmes, S., Lamarque, J. F., Jimenez, J. L.,
    Campuzano-Jost, P., Nault, B. A., Crounse, J. D., Wennberg, P. O., Kim, M., Allen, H., Ryerson,
70      T. B., Thompson, C. R., Peischl, J., Moore, F., Nance, D., Hall, B., Elkins, J., Tanner, D., Huey,
    L. G., Hall, S. R., Ullmann, K., Orlando, J. J., Tyndall, G. S., Flocke, F. M., Ray, E., Hanisco,
    T. F., Wolfe, G. M., St. Clair, J., Commane, R., Daube, B., Barletta, B., Blake, D. R., Weinzierl,
    B., Dollner, M., Conley, A., Vitt, F., Wofsy, S. C., Riemer, D. D., and Apel, E. C.: Atmospheric
    Acetaldehyde: Importance of Air-Sea Exchange and a Missing Source in the Remote
75      Troposphere, Geophys. Res. Lett., 46, 5601-5613, http://doi.org/10.1029/2019gl082034, 2019.
Wu, T., Zhang, F., Zhang, J., Jie, W., Zhang, Y., Wu, F., Li, L., Yan, J., Liu, X., Lu, X., Tan, H.,
    Zhang, L., Wang, J., and Hu, A.: Beijing Climate Center Earth System Model version 1 (BCC-
    ESM1): model description and evaluation of aerosol simulations, Geoscientific Model
    Development, 13, 977-1005, http://doi.org/10.5194/gmd-13-977-2020, 2020.

80

**Comment#1-3:** Figure 6: The y-Axis is unlabeled and not clearly explained in the
caption. I assume this is ozone in ppb, but some label or caption edit is probably in
order. More importantly, it is not clear from the discussion of Figure 6 and Figure 3
85  whether the simulated values are taken from GEOS-Chem over some defined region or
simply from the grid boxes corresponding to the observations in Figure 3.
**Response#1-3: We have added the label "Ozone mixing ratio [ppbv]" for the y-
axis in Figure 6. We now also state in Section 3.1 (Observations used for model**

evaluation): "**To derive the monthly mean ozone profiles, only sites and months with more than three observations per month are considered, and simulated monthly mean ozone profiles are sampled over the corresponding model grids (Lu et al., 2019b).**"

Reference added:

Lu, X., Zhang, L., Zhao, Y., Jacob, D. J., Hu, Y., Hu, L., Gao, M., Liu, X., Petropavlovskikh, I., McClure-Begley, A., and Querel, R.: Surface and tropospheric ozone trends in the Southern Hemisphere since 1990: possible linkages to poleward expansion of the Hadley circulation, Science Bulletin, 64, 400-409, http://doi.org/10.1016/j.scib.2018.12.021, 2019b.

**Comment#1-4:** I have questions about the choices of regions used in Figures 3 and 6, though I recognize the authors are referencing regions already in the literature. In particular, I do not agree with the classification of the "NH Low Latitude" grouping in the context of this paper: lumping together southern China, Hawaii, Panama, French Guiana, and Nigeria into one coherent region would to elide an enormous amount of difference in chemical regimes, biomass burning, and anthropogenic emissions between the various regions chosen. I believe it makes more sense to separate the points near China and Taiwan into a "Southeast Asia" region, separate from the remaining "NH Low Latitude" grouping. As the "SH midlatitude" region already has an N of 2, this should be similarly acceptable. This choice may have ramifications for the seasonal simulation performance as summarized in Figure 6.

**Response#1-4: Thanks for pointing it out. We have followed the reviewer's suggestion and separated the original "NH Low Latitude" group into the "Southeast Asia" and "NH Low Latitude" groups. Figures 3, 5, and 6 are re-plotted and they do not affect our analysis.**

**Comment#1-5:** Figure 7: The choice made to show observations as small circles on a global map is very confusing in this context. Given the density of spatial coverage over the Northeastern US and Europe, I suggest 3 possible fixes. 1) grid the observations to the same resolution as the model output and show them side-by-side with the simulated ozone; 2) show different plots for the US, for Europe, and for the rest of the globe if necessary; or 3) the points should be averaged for display purposes over a larger spatial extents and made larger themselves. The currently displayed global view could be placed in the SI. As of now, Figure 7 serves a mostly pro forma purpose – it is hard for me to glean any useful information from the plot given the display, and true patterns in the observation data (for example the increase in surface ozone in the observations over central Europe) are obscured by the chosen plotting scheme, rather than illustrated. If the authors do not believe that Figure 7 should be changed, perhaps it belongs in the SI in its entirety, as it is not clear to me what the figure adds to the paper in its current state.

**Response#1-5: We agree. We have revised Figure 7 as attached below to show a side-by-side comparison of simulated and observed surface ozone at the same model grids. This change does not affect our analysis.**

[Figure]

**Figure 7.** Spatial and seasonal distributions of observed and simulated surface ozone mixing ratios over 2012-2014. The model results (right panels) are compared to observations at rural/remote sites from the TOAR dataset (left panels). Observations are averaged to the same model grid. Seasonal mean values for observations and model results, their spatial correlation coefficients (*r*), and the number of co-sampled grids (N) are shown inset.

**Comment#1-6:** Figure 8: This is an excellent figure – very clear, with a high information-to-ink ratio. One comment: it seems unlikely that the geographic equator is a meaningful division in this kind of global average, given that the 'meteorological equator' deviates from 0 degrees depending upon season and region. If the comparison datasets support the option, the authors should consider using 5 latitude bins instead of 4 – perhaps 90-50S, 50-20 S, 20 S - 20 N, 20-50 N, and 50-90 N? This has the benefit of treating the tropics, midlatitudes, and polar-latitudes differently.

**Response#1-6: Thank you for the nice words on the figure. We agree that using latitude bins of 90-50S, 50-20S, 20S-20N, 20-50N, and 50-90N can be better. However, the results of Young et al. (2013) and Emmons et al. (2010) are presented as the given latitude averages and we followed here to compare with their results.**

**Reference:**

Emmons, L. K., Walters, S., Hess, P. G., Lamarque, J. F., Pfister, G. G., Fillmore, D., Granier, C.,

Guenther, A., Kinnison, D., Laepple, T., Orlando, J., Tie, X., Tyndall, G., Wiedinmyer, C., Baughcum, S. L., and Kloster, S.: Description and evaluation of the Model for Ozone and Related chemical Tracers, version 4 (MOZART-4), Geoscientific Model Development, 3, 43-67, http://doi.org/10.5194/gmd-3-43-2010, 2010.

Young, P. J., Archibald, A. T., Bowman, K. W., Lamarque, J. F., Naik, V., Stevenson, D. S., Tilmes, S., Voulgarakis, A., Wild, O., Bergmann, D., Cameron-Smith, P., Cionni, I., Collins, W. J., Dalsøren, S. B., Doherty, R. M., Eyring, V., Faluvegi, G., Horowitz, L. W., Josse, B., Lee, Y. H., MacKenzie, I. A., Nagashima, T., Plummer, D. A., Righi, M., Rumbold, S. T., Skeie, R. B., Shindell, D. T., Strode, S. A., Sudo, K., Szopa, S., and Zeng, G.: Pre-industrial to end 21st century projections of tropospheric ozone from the Atmospheric Chemistry and Climate Model Intercomparison Project (ACCMIP), Atmos. Chem. Phys., 13, 2063-2090, http://doi.org/10.5194/acp-13-2063-2013, 2013.

**Comment#1-7:** Figure 10: why is the discrepancy between modeled and observed SO2 not discussed further? It is true that the model does broadly reproduce the spatial trend observed over China and India but elsewhere the correlation would appear to be quite poor.

**Response#1-7: Thanks for pointing it out. We have partly reduced the discrepancies between the observed and modelled SO$_2$, by removing OMI measurements with slant columns greater than 5 Dobson Units (1.34 × 10$^{17}$ molecules cm$^{-2}$) which are affected by strong eruptive volcanoes (Lee et al., 2009, 2011).**

**We further state in Section 3.4 (Evaluation of other atmospheric constituents) "We find low biases in the modelled PBL SO$_2$ especially over the volcanic eruption regions (e.g., Central Africa) but high biases in the industrialized regions such as East Asia, a pattern consistent with previous comparisons between the OMI and GEOS-Chem PBL SO$_2$ columns, which may reflect inappropriate ship and volcanic emissions in the model (Lee et al., 2009) and/or the model bias in the PBL height."**

**Reference added:**

Lee, C., Martin, R. V., van Donkelaar, A., O'Byrne, G., Krotkov, N., Richter, A., Huey, L. G., and Holloway, J. S.: Retrieval of vertical columns of sulfur dioxide from SCIAMACHY and OMI: Air mass factor algorithm development, validation, and error analysis, J. Geophys. Res., 114, http://doi.org/10.1029/2009jd012123, 2009.

Lee, C., Martin, R. V., van Donkelaar, A., Lee, H., Dickerson, R. R., Hains, J. C., Krotkov, N., Richter, A., Vinnikov, K., and Schwab, J. J.: SO2emissions and lifetimes: Estimates from inverse modeling using in situ and global, space-based (SCIAMACHY and OMI) observations, J. Geophys. Res., 116, http://doi.org/10.1029/2010jd014758, 2011.

**Comment#1-8:** Technical Corrections: Line 448: Citation is misspelled – should be "Kodros and Pierce, 2017". The reference is correctly spelled in the bibliography.

**Response#1-8: Corrected.**

---

## Author Comment (AC2) · 10 Jul 2020

**Reply of RC2: Review of Lu et al., "Development of the global atmospheric general circulation-chemistry model BCC-GEOS-Chem v1.0: model description and evaluation"**

**Reviewer #2**

**Comment#2-1:** The manuscript presents an overview and assessment of a newly constructed chemistry-climate model that has resulted from the linking of GEOS-Chem with the Beijing Climate Center AGCM. The general features of the model are presented and a fairly extensive comparison against observations for ozone are presented. In addition, other aspects of the model chemical climate are presented, including the global distribution of OH, ozone budget terms, and some limited comparisons for aerosol quantities against observations including AOD and speciated aerosol concentrations over the US.

The manuscript is very clearly written and presented, providing a fairly complete overview of the model components and an idea of how the chemical climate of BCC GEOS-Chem compares with GEOS-Chem itself and with other chemistry climate models.

**Response#2-1: We thank the reviewer for the valuable comments. All of them have been implemented in the revised manuscript. Please see our itemized responses below.**

**Comment#2-2:** My only significant criticism is focused on Figure 10, comparing column amounts of NO2, CO, SO2 and CH2O. In the discussion of Figure 10, lines 394 – 404, the authors say that the averaging kernels for the satellite observations were not applied to the model concentrations when calculating the column amounts and that the comparisons 'mainly focus ... on spatial variations rather than absolute magnitudes.' In this case the comparison is nothing more than a test to make sure the specified emissions are being put into the model in the correct locations. Global models have a long-standing low bias for CO in the northern hemisphere that appears to be related to emissions and the hydroxy radical, but Figure 10 shows that BCC-GEOS-Chem has too high CO in the Northern Hemisphere. Due to the lack of a quantitative comparison with the satellite data by application of the averaging kernel it is impossible to judge whether the differences signify anything. While CH2O is not predominantly due to direct emission, the spatial distribution is tightly coupled to the emissions of biogenic hydrocarbons, so the comparison will also be largely driven by having regions of high biogenic emissions in the correct place. The differences in the magnitude of CH2O between the satellite and model is quite large and it would be interesting to have a more quantitative comparison with the satellite observations as the qualitative comparison focused on the spatial distribution is not informative at all. I would strongly urge the authors to revise the comparison of the column amounts to be more quantitative by application of the appropriate averaging kernel. My other comments are all minor in nature and are given below.

**Response#2-2: We agree with the reviewer on the need to improve the comparison and discussion for these chemical constituents. For CO, we have now applied the satellite averaging kernels to smooth the model results, and focus on the comparison at 700 hPa where MOPITT has generally high sensitivities. As shown in the new Fig. 10a, the large model high bias has been significantly improved though still exists. For other constituents, however, we do not apply corresponding averaging kernel, which requires additional 3-year model simulation to co-sample observations along the satellite tracks. Previous studies (Zhu et al., 2016; 2020) showed $CH_2O$ shape factor (a prior) was not the main driver of the discrepancy between GEOS-Chem modeled and retrieved columns. Their studies also revealed that satellite $CH_2O$ retrievals showed significant low bias (up to 50%) compared to aircraft measurements, which may largely explain the model high bias as shown here. For $SO_2$, we have partly reduced the discrepancies between the observed and modelled $SO_2$, by removing OMI measurements with slant columns greater than 5 Dobson Units ($1.34 \times 10^{17}$ molecules $cm^{-2}$) which are affected by strong eruptive volcanoes (Lee et al., 2009, 2011).**

**We have added much more discussions in Section 3.4 (Evaluation of other atmospheric constituents)**
**"Figure 10 compares the spatial distributions of annual mean simulated CO, $NO_2$, $SO_2$, and $CH_2O$ with satellite observations. We evaluate CO at 700 hPa where MOPITT satellite has generally high sensitivity (Emmons et al., 2004; Pfister et al., 2005), and apply averaging kernel to smooth the modelled CO. As shown in Figure 10, BCC-GEOS-Chem v1.0 reproduces the high CO levels over the northern mid-latitudes driven by high anthropogenic sources, and over the central Africa driven by biomass burning emissions (spatial correlation coefficient $r=0.92$) with some overestimates. It also captures the observed hotspots of tropospheric $NO_2$ ($r=0.87$) and PBL $SO_2$ columns ($r=0.32$) over East Asia that generally follow the distribution of anthropogenic sources. The sharp land-ocean gradients for both tracers reflect their short chemical lifetime. We find low biases in the modelled PBL $SO_2$ especially over the volcanic eruption regions (e.g., Central Africa) but high biases in the industrialized regions such as East Asia, a pattern consistent with previous comparisons between the OMI and GEOS-Chem PBL SO2 columns, which may reflect inappropriate ship and volcanic emissions in the model (Lee et al., 2009) and/or the model bias in the PBL height. High levels of tropospheric $CH_2O$ column are simulated over the Amazon, the central Africa, tropical Asia, and the southeastern US, where $CH_2O$ oxidized from large biogenic emissions of VOCs ($r=0.67$), but the model shows notable overestimates. Previous studies (Zhu et al., 2016; 2020) showed that satellite $CH_2O$ retrievals are biased low by 20–51% compared to aircraft measurements which would partly explain the model bias. Future assessments are required to correct the biases of these gaseous pollutants."**

[Figure]

**Figure 10.** Spatial distributions of satellite observed (top panels) and model simulated (bottom panels) annual mean (a) CO mixing ratio at 700 hPa, (b) tropospheric NO₂ column, (c) SO₂ column in planetary boundary layer, and (d) tropospheric CH₂O column,. Values are 3-year averages for 2012-2014.

**References added:**

Emmons, L. K., Deeter, M. N., Gille, J. C., Edwards, D. P., Attié, J. L., Warner, J., Ziskin, D., Francis, G., Khattatov, B., Yudin, V., Lamarque, J. F., Ho, S. P., Mao, D., Chen, J. S., Drummond, J., Novelli, P., Sachse, G., Coffey, M. T., Hannigan, J. W., Gerbig, C., Kawakami, S., Kondo, Y., Takegawa, N., Schlager, H., Baehr, J., and Ziereis, H.: Validation of Measurements of Pollution in the Troposphere (MOPITT) CO retrievals with aircraft in situ profiles, J. Geophys. Res., 109, n/a-n/a, http://doi.org/10.1029/2003jd004101, 2004.

Lee, C., Martin, R. V., van Donkelaar, A., O'Byrne, G., Krotkov, N., Richter, A., Huey, L. G., and Holloway, J. S.: Retrieval of vertical columns of sulfur dioxide from SCIAMACHY and OMI: Air mass factor algorithm development, validation, and error analysis, J. Geophys. Res., 114, http://doi.org/10.1029/2009jd012123, 2009.

Lee, C., Martin, R. V., van Donkelaar, A., Lee, H., Dickerson, R. R., Hains, J. C., Krotkov, N., Richter, A., Vinnikov, K., and Schwab, J. J.: SO2emissions and lifetimes: Estimates from inverse modeling using in situ and global, space-based (SCIAMACHY and OMI) observations, J. Geophys. Res., 116, http://doi.org/10.1029/2010jd014758, 2011.

Pfister, G., Hess, P. G., Emmons, L. K., Lamarque, J. F., Wiedinmyer, C., Edwards, D. P., Petron, G., Gille, J. C., and Sachse, G. W.: Quantifying CO emissions from the 2004 Alaskan wildfires using MOPITT CO data, Geophys. Res. Lett., 32, http://doi.org/Artn L1180910.1029/2005gl022995, 2005.

Zhu, L., Jacob, D. J., Kim, P. S., Fisher, J. A., Yu, K., Travis, K. R., Mickley, L. J., Yantosca, R. M., Sulprizio, M. P., De Smedt, I., Abad, G. G., Chance, K., Li, C., Ferrare, R., Fried, A., Hair, J. W., Hanisco, T. F., Richter, D., Scarino, A. J., Walega, J., Weibring, P., and Wolfe, G. M.: Observing atmospheric formaldehyde (HCHO) from space: validation and intercomparison of six retrievals from four satellites (OMI, GOME2A, GOME2B, OMPS) with SEAC(4)RS aircraft observations over the Southeast US, Atmos. Chem. Phys., 16, 13477-13490, http://doi.org/10.5194/acp-16-13477-2016, 2016.

Zhu, L., González Abad, G., Nowlan, C. R., Chan Miller, C., Chance, K., Apel, E. C., DiGangi, J. P., Fried, A., Hanisco, T. F., Hornbrook, R. S., Hu, L., Kaiser, J., Keutsch, F. N., Permar, W., St. Clair, J. M., and Wolfe, G. M.: Validation of satellite formaldehyde (HCHO) retrievals using observations from 12 aircraft campaigns, Atmospheric Chemistry and Physics Discussions,

http://doi.org/10.5194/acp-2019-1117, 2020.

**Comment#2-3:** Lines 80 - 83: Here the authors state 'Integration of GEOS-Chem chemical module into CSMs has been enabled by separating the module (which simulates all local processes including chemistry, deposition, and emission) from the simulation of transport, and making it operate on 1-D (vertical) columns in a grid-independent manner (Long et al., 2015; Eastham et al., 2018).' How is the 1-D column version of GEOS-Chem integrated with a 3-D CSM for processes that typically occur in the physics of the model such as vertical turbulent diffusion and transport by deep convection? (I do find a description of deep convection and wet deposition around line 190, but no mention of how vertical diffusion is performed.)

**Response#2-3: Thanks for pointing it out. The vertical diffusion of the tracers is parameterized using a non-local scheme as described in Holtslag and Boville (1993). We now state in the Section 2.2 (Atmospheric Chemistry): "Tracer advection in BCC-GEOS-Chem v1.0 is performed using a semi-Lagrangian scheme (Williamson and Rasch, 1989) and the vertical diffusion within the boundary layer follows the parameterization of Holtslag and Boville (1993)"**

**Reference added:**

Holtslag, A. A. M. and Boville, B. A.: Local versus nonlocal boundary-layer diffusion in a global climate model, J. Climate, 6, 1825–1842, 1993.

**Comment#2-4:** Lines 181 – 183: The dry deposition uses the general characteristics of the land surface as given by the CSM land module BCC-AVIM. Are there also links to the land surface scheme for more short-term variables such as stomatal resistance, that would allow for effects such as drought on dry deposition?

**Response#2-4: The stomatal conductance is simulated but its influences on dry deposition are not considered, therefore the model does not allow drought influences the dry deposition through modulating stomatal conductance so far. We now state in Section 2.5 (Dry and wet deposition) "Variables needed for the dry deposition calculation such as the friction velocity, Monin-Obukhov length, and leaf area index (LAI) are obtained from the atmospheric dynamics/physics modules or the land module BCC-AVIM, based on which GEOS-Chem calculates the aerodynamic, boundary-layer, and surface resistances. The impacts of some other short-term land variables, such as stomatal conductance, on dry deposition are not included yet."**

**Comment#2-5:** Line 234: Minor typo in 'The model estimates t global annual ...'
**Response#2-5: Corrected.**

**Comment#2-6:** Lines 311 – 314: Somewhere, either in the discussion of Figure 5 or the caption, there should be mention that the comparison is for annual average ozone.
**Response#2-6: Thanks for pointing it out. We now state "As shown in Figure 5,**

**the model well reproduces the observed annual mean ozone vertical structures…”.**
**We have also revised the figure caption accordingly.**

**Comment#2-7:** Lines 314 – 321: I was a bit curious about why the vertical profile of ozone for the Japanese stations shows such a different vertical structure between the observations and model in Figure 5. Looking at Figure 6, the 300 hPa doesn't show that big of a difference. If 300 hPa is somewhere around 10 – 11 km, shouldn't the annual average in the observations be over 120 ppbv, though it is listed as 90 ppbv on Figure 6?

**Response#2-7: 300hPa over Japan is around 9 km, and therefore Figures 5 and 6 are consistent. This is a region with frequent stratosphere to troposphere transport and the model may have difficulty in capturing the sharp ozone increases with increasing altitude there.**

**Comment#2-8:** Line 364: Discussing the discrepancy in OH in the tropics between the Spivakovsky climatology the authors state 'This discrepancy appears to be mainly driven by the high bias in ozone levels in this region.' Attempts to understand the reasons for differences in OH between models has shown how many different factors play a role – see, for example, Nicely et al. Atmos. Chem. Phys. 20, doi:10.5194/acp-20-1341-2020, 2020. Do the authors have some reason to believe that the ozone and hydroxyl biases are related and, if not I would suggest removing this statement.

**Response#2-8: Thanks for pointing it out. We agree that the OH discrepancies in the model can be driven by more factors than the ozone bias. We now change the statement to “Discrepancies in modeling climate and concentrations of methane, ozone, NOx, and CO can all contribute to the OH bias in climate-chemistry models (Nicely et al., 2020).”**

**Reference added:**

Nicely, J. M., Duncan, B. N., Hanisco, T. F., Wolfe, G. M., Salawitch, R. J., Deushi, M., Haslerud, A. S., Jöckel, P., Josse, B., Kinnison, D. E., Klekociuk, A., Manyin, M. E., Marécal, V., Morgenstern, O., Murray, L. T., Myhre, G., Oman, L. D., Pitari, G., Pozzer, A., Quaglia, I., Revell, L. E., Rozanov, E., Stenke, A., Stone, K., Strahan, S., Tilmes, S., Tost, H., Westervelt, D. M., and Zeng, G.: A machine learning examination of hydroxyl radical differences among model simulations for CCMI-1, Atmos. Chem. Phys., 20, 1341-1361, http://doi.org/10.5194/acp-20-1341-2020, 2020.